# Close to Reality: Interpretable and Feasible Data Augmentation for Imbalanced Learning

## Abstract

Many machine learning classification tasks involve imbalanced datasets, which are often subject to over-sampling techniques aimed at improving model performance. However, these techniques are prone to generating unrealistic or infeasible samples. Furthermore, they often function as black boxes, lacking interpretability in their procedures. This opacity makes it difficult to track their effectiveness and provide necessary adjustments, and they may ultimately fail to yield significant performance improvements. To bridge this gap, we introduce the Decision Predicate Graphs for Data Augmentation (DPG-da), a framework that extracts interpretable decision predicates from trained models to capture domain rules and enforce them during sample generation. This design ensures that over-sampled data remain diverse, constraint-satisfying, and interpretable. In experiments on synthetic and real-world benchmark datasets, DPG-da consistently improves classification performance over traditional over-sampling methods, while guaranteeing logical validity and offering clear, interpretable explanations of the over-sampled data.

## 1 Introduction

A recurring challenge in supervised classification is the presence of imbalanced datasets, where one or more classes are represented by significantly fewer samples than others. This imbalance can significantly create biases in the learning algorithms toward the majority class, leading to poor generalization on minority classes. As a result, classifiers may achieve high overall accuracy while systematically misclassifying rare but critical events, a limitation that is particularly problematic in high-stakes domains such as fraud detection (Makki et al., 2019; Rubaidi et al., 2022; Silva et al., 2023) and clinical diagnosis (Roy et al., 2024; Ejiyi et al., 2025), where correct classification of the minority class is often the central goal.

Existing approaches to class imbalance, focusing on tabular data, fall into three categories: problem definition, algorithmic modification, and data-level processing (He & Ma, 2013; Weiss, 2013; Haixiang et al., 2017). The first reformulates evaluation metrics or tasks to emphasize minority performance but does not alter the data or model. The second, often referred to as cost-sensitive or cost-adjusted learning, integrates class-specific misclassification penalties directly into the loss function (Cao et al., 2013; Hu et al., 2017), but its effectiveness hinges on defining appropriate costs and often results in more complex models. The third, and most widely adopted, modifies the data distribution directly, either by under-sampling the majority class or over-sampling the minority class. Under-sampling, as in Random Under-Sampling (RUS) (Weiss, 2013), risks discarding informative instances (He & Ma, 2013), while over-sampling techniques like Random Over-Sampling (ROS) (Chawla et al., 2002), which simply duplicate existing minority instances, may induce overfitting (Dixit et al., 2023).

More sophisticated over-sampling methods, such as the interpolation-based SMOTE (Chawla et al., 2002) and generative approaches like GANs (Goodfellow et al., 2020) and VAEs (Xu et al., 2019), aim to generate diverse synthetic samples. However, these methods face three key limitations: (i) interpolation-based techniques can generate infeasible instances that violate domain constraints (Matharaarachchi et al., 2024; Yang et al., 2023), (ii) generative models may suffer from instability and provide no guarantees of logical validity (Di Liello et al.,

2020; Tarawneh et al., 2022), and (iii) both approaches offer little transparency into how synthetic samples are produced, limiting their usefulness in sensitive domains.

These issues are particularly acute in high-stakes applications, where synthetic data must not only improve classifier performance but also remain valid, auditable, and interpretable. Interpretability, in contrast to post-hoc explainability, refers to methods whose internal logic and outputs can be directly understood without approximation layers (Ross & Doshi-Velez, 2018; Rudin, 2019). In the context of data augmentation, the process of generating synthetic samples from a query sample to expand or balance existing datasets, interpretability requires not only that the generated records respect domain-specific constraints, but also that users gain insight into how these synthetic samples are produced and in what ways they differ from the original data they are derived from.

To address these challenges, we propose *DPG-da*, a constraint-aware augmentation method guided by Decision Predicate Graphs (DPGs) (Arrighi et al., 2024). DPGs provide a global, interpretable representation of decision logic, from which class-specific predicates are extracted and enforced during generation. **As currently formulated, DPG-da operates on tabular data, since the decision predicates are defined as coordinate-wise thresholds on feature values.** In our framework, interpretability arises in two complementary ways: (i) constraints define a transparent and auditable valid region for minority augmentation, offering users an explicit understanding of the problem space; and (ii) the evolutionary transformation process from query samples to synthetic samples makes it possible to trace which features changed most during augmentation, thereby explaining *how* new instances were created.

The contributions of this work are:

- A novel augmentation framework, DPG-da, that integrates constraint discovery and enforcement directly into sample generation, ensuring validity and interpretability of synthetic data.

- A systematic violation analysis across three complementary groups of datasets (handcrafted rule-based, high-stakes real-world, and synthetic overlapping) and over 10 baseline over-samplers, showing that existing interpolation- and generative-based methods frequently produce infeasible samples, whereas DPG-da guarantees strict constraint adherence.

- A comprehensive performance evaluation on 27 benchmark datasets, demonstrating that DPG-da consistently outperforms interpolation and generative over-sampling methods in classification performance while avoiding violations and providing interpretability into the augmentation process.

Experiments confirm that DPG-da outperforms state-of-the-art over-sampling techniques in both accuracy and reliability, while eliminating infeasible samples. Although this comes with higher computational cost, the trade-off is justified in domains where validity and interpretability of augmented data are as critical as predictive accuracy.

The remainder of this paper is organized as follows: Section 2 provides the necessary background, Section 3 reviews related work, Section 4 presents the DPG-da framework, Section 5 details the experimental design, Section 6 reports results and discussion, and Section 8 concludes with contributions and future directions.

## 2 Background

### 2.1 Violation Space

Let $\mathcal{X}$ denote the problem space, i.e., all possible combinations of $x$ feature values (e.g., $\mathcal{X} \subseteq \mathbb{R}^d$ for $d$ features). Not every point in $\mathcal{X}$ represents a feasible or meaningful instance in reality.

The *violation space* $V \subseteq \mathcal{X}$ is the subset of points that break one or more domain-specific constraints, such as physical laws, regulatory requirements, or common-sense rules. For example, a record describing a patient with negative blood pressure, or a loan applicant under 18 years old, belongs to $V$. These samples are numerically valid but semantically invalid.

Formally, let $\mathcal{C}$ be the set of Boolean-valued constraints over $\mathcal{X}$. Then:

$$V = \{x \in \mathcal{X} \mid \exists c \in \mathcal{C} : c(x) = \text{False}\}.$$

Its complement, the *feasible region*, consists of instances that satisfy all constraints. The literature sometimes refers to $V$ as the *infeasible region* (Peres et al., 2018; Alsouly et al., 2022) or the *constraint violation region* (Bernardo et al., 2011).

## 2.2 Decision Predicate Graphs

DPGs (Arrighi et al., 2024) are graph-based structures that capture decision logic in an interpretable and compact form. A predicate $p = (f, o, t)$ encodes a simple test over a feature $f$ with operator $o \in \{<, \leq, >, \geq\}$ and threshold $t \in \mathbb{R}$. Nodes correspond to predicates or leaf outcomes; edges connect predicates that co-occur along decision paths, with weights indicating their frequency. Figure 1 shows an illustrative binary-class example.

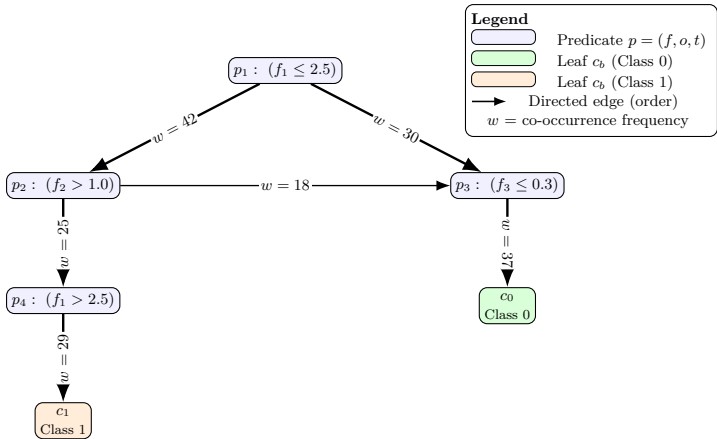

Figure 1: DPG structure showing predicates as nodes and class leaves.

DPGs are typically extracted from surrogate ensemble models trained to approximate a base classifier. Formally, a DPG for model $M_n$ is a directed weighted graph $\text{DPG}(M_n) = (P, E)$, where $P$ is the set of predicates and $E$ the edges weighted by co-occurrence frequency. Leaf nodes $c_b \in P$ denote final class labels.

From a DPG, one can derive *class-bound predicates*, i.e., conjunctions of rules that consistently lead to a given class. These yield explicit descriptions of feasible regions, such as $18 \leq \text{Age} \leq 65$ (Loan requirement for approval) or $\texttt{Glucose} < 140$ (Diabetes diagnosis). In our setting, these constraints serve two roles: they restrict augmentation to the feasible region, preventing violations, and they provide interpretable insights into the structure of class-specific domains.

## 3 Related Works

Data augmentation for tabular data has been widely studied, primarily to mitigate class imbalance. Early over-sampling methods such as SMOTE and its variants (Han et al., 2005; Kovács, 2019) interpolate between minority instances, while generative models such as GANs and VAEs (Ding et al., 2024; Ai et al., 2023; Tian et al., 2024; Stocksieker et al., 2024; Yadav et al., 2025; Berti et al., 2025) attempt to learn full data distributions. Although effective at enlarging datasets, both families often yield implausible or noisy samples, as they lack mechanisms to enforce domain-specific constraints and operate largely as black-box processes (Thekumparampil et al., 2018; Sakho et al., 2024).

To improve trust and transparency, researchers have paired augmentation with XAI. Prior work combined SMOTE with post-hoc explanation tools such as SHAP, LIME, or LRP in domains including fraud detection, diabetes prediction, and student performance forecasting (Patil et al., 2020; Nayan et al., 2023;

Sahlaoui et al., 2024). While these studies demonstrate the value of combining over-sampling with interpretability, augmentation itself remains unconstrained and prone to producing unrealistic samples. More recent domain-specific frameworks embed interpretability directly into augmentation pipelines, for example using model explanations to guide low-resource NLP augmentation (Mersha et al., 2025) or steering over-sampling with feature-importance signals in healthcare (Ejiyi et al., 2025). These methods, however, remain tied to particular data modalities.

A related line of work employs counterfactual explanations to generate locally plausible alternatives (Mohammed et al., 2022; Temraz & Keane, 2022; Panagiotou et al., 2024). Counterfactual augmentation improves minority coverage and interpretability, but is inherently instance-specific, sensitive to learned decision boundaries, and not designed for large-scale rebalancing. Manifold-based approaches such as TabMDA (Margeloiu et al., 2024) address augmentation at the dataset level by leveraging pre-trained tabular models, but they lack transparency and do not enforce domain-specific constraints, limiting their applicability in sensitive domains.

In summary, existing methods trade off between diversity (e.g., generative, manifold) and plausibility or interpretability (e.g., counterfactuals, XAI-assisted over-sampling). Our framework, DPG-da, bridges this gap by embedding model-derived constraints directly into the augmentation process, ensuring that generated samples are both semantically valid and interpretable.

## 4   Proposed Approach

DPG-da consists of two key components: (i) extraction of class-specific constraints using Decision Predicate Graphs (DPGs) and (ii) augmentation of minority samples via a heuristic optimization process of a query sample. The framework ensures that generated data points are both diverse and strictly confined to feasible regions defined by the DPG, addressing limitations of SMOTE-based and generative approaches that may produce infeasible or overlapping samples (Figure 2).

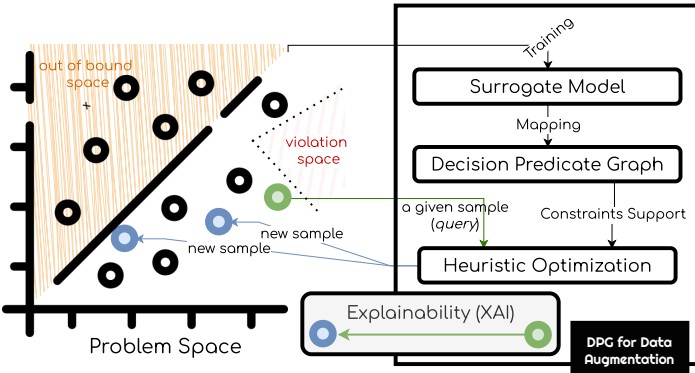

Figure 2: Workflow of the proposed DPG-da. Constraints extracted from DPGs guide the optimization for valid and diverse synthetic samples.

### 4.1   Surrogate Model

The original imbalanced dataset is first used to train a Random Forest (RF), which serves as a surrogate model approximating the decision boundaries of the base classifier. The surrogate enables the systematic extraction of decision predicates, which are subsequently used to construct the Decision Predicate Graph (DPG). These predicates define class-specific constraints that delimit feasible regions for synthetic sample generation, thereby ensuring interpretability through explicit, data-driven rules.

While we adopt RFs in our implementation, DPG-da does not require RFs per se. This choice is motivated by two practical advantages. First, RFs naturally produce interpretable, axis-aligned predicate rules through decision-tree splits, which can be directly encoded as nodes and edges in a DPG. Second, feasibility checks and distances to violated predicates can be computed efficiently, making RFs well suited for the constraint evaluation and fitness computation steps of the augmentation process.

More generally, DPG-da only requires a surrogate model that satisfies the following properties: (i) it yields an explicit set of predicates or constraints defining class-consistent regions of the feature space, such that these constraints can be represented within a DPG; and (ii) it allows efficient evaluation of predicate satisfaction, and optionally a distance-to-predicate measure used in the evolutionary fitness function.

Under these conditions, alternative surrogate models are compatible with the framework, including gradient-boosted decision trees, rule-based models (e.g., rule lists or rule sets), or oblique decision trees that produce linear predicates.

## 4.2 Heuristic Optimization

To generate augmented samples confined to the valid class-bound regions (*constraints*) identified by the DPG, we employ a heuristic optimization strategy based on Genetic Algorithms (GAs). Each augmentation begins from a query or seed sample $x^q$ drawn from the minority class, which serves as a reference for generating new candidates. We selected GAs for their simplicity and robustness: although faster or more sophisticated optimization methods exist, our focus is on constraint-guided augmentation rather than optimizing the search process itself.

Each individual in the GA represents a candidate augmented sample $x^a$, initialized by uniform sampling within the DPG-defined bounds for the target class. Candidates are evaluated using a composite fitness function that captures multiple objectives:

- **Classification validity** $V(x^a, x^q)$: A binary indicator that evaluates whether the candidate sample $x^a$ and the query sample $x^q$ are assigned to the same target class, thereby validating semantic alignment with the intended class label.

- **DPG-constraint adherence** $A(x^a, P)$: Rewards candidate samples that satisfy class-specific predicates, remaining strictly within the feasible region defined by the DPG class bounds. This prevents over-generalization and preserves the structure of the original data manifold.

- **Distance from the query sample** $D(x^a, x^q)$: Promotes diversity among the augmented samples by maximizing the distance between each new augmented sample and the reference query sample. Distance is measured using Euclidean distance for continuous features.

- **Augmentation sparsity** $S(x^a, x^q)$: Encourages minimal and interpretable modifications by penalizing candidates that differ significantly from the query sample. Sparsity is quantified using the $\ell_0$ norm, which counts the number of feature changes, thereby promoting localized, meaningful transformations.

The overall fitness of a candidate $x^a$, with respect to the query sample $x^q$ and the class-bound predicates $P$ (obtained from the DPG), is computed by weighting the contributions of multiple objectives. Specifically, the fitness is defined as:

$$\text{fitness}(x^a, x^q, P) = V(x^a, x^q) \times \big( w_1 \cdot A(x^a, P) + w_2 \cdot D(x^a, x^q) + w_3 \cdot (1 - S(x^a, x^q)) \big) \tag{1}$$

where $w_1, w_2, w_3$ control the relative importance of each objective. Higher fitness scores guide the GA toward candidates that best satisfy all objectives.

The GA iterates until a stopping criterion is met: either the best fitness plateaus for a fixed number of generations or a maximum number of generations is reached. This ensures efficient exploration of feasible regions while generating diverse and interpretable samples without excessive computational overhead.

### 4.3 Interpretability of Augmentation

Our proposed DPG-da method allows traceability of the augmentation process, while also providing a limited form of interpretability of the data structure. Specifically, by tracking the evolutionary trajectory of each augmented sample produced via the GA, users can follow how candidate samples change over generations and verify constraint adherence. This traceability does not fully explain why a sample is valid in a causal sense, but it enables post-hoc validation of synthetic data feasibility and inspection of feature dynamics.

To facilitate analysis, we generate feature-level visualizations (e.g., heatmaps and barplots) that show the magnitude and direction of changes for the best-performing individuals at each generation. This is inspired by XAI approaches that highlight feature importance or dynamics in other domains (Richard et al., 2024; Sawada & Toyoda, 2019).

For example, consider a binary loan approval classification problem with four numerical features: $x_1$ (Age in years), $x_2$ (Annual Income in thousands of USD), $x_3$ (Credit Score ranging from 300 to 850), and $x_4$ (Number of Children). Let $\mathbf{x}^{(0)} = [45, 50, 620, 2]$ be a query instance from the minority class (*approved loans*). The DPG derived from the surrogate model defines the feasible region for synthetic instances through the following constraints: $x_1 \geq 30$, $x_2 \geq 45$, $x_3 \geq 600$, and $x_4 \leq 3$. The goal of DPG-da is to generate synthetic instances $\mathbf{x}^{(g)}$ at each generation $g$ of the GA while respecting these constraints.

In this example, the evolutionary process produces the best individuals over three generations: $\mathbf{x}^{(1)} = [47, 52, 610, 2]$, $\mathbf{x}^{(2)} = [50, 55, 605, 3]$, and $\mathbf{x}^{(3)} = [52, 60, 590, 3]$, with the last instance violating constraint $C_3$. The algorithm adjusts $x_1$ and $x_2$ to explore the feasible region and promote diversity, while the violation of $C_3$ results in penalization and exclusion. Tracking these changes demonstrates the traceability of the GA-based augmentation, while the constraints themselves provide a limited interpretability of the data structure.

The change in feature values between successive generations is shown in Table 1, illustrating how the GA explores the feasible space while enforcing constraints.

Table 1: Feature-wise changes ($\Delta x_i$) between successive generations of the GA in the DPG-da augmentation process. A violation of constraint $C_3$ occurs in generation 3 (signed by $\times$).

| Feature | $\mathbf{\Delta^{0 \to 1}}$ | $\mathbf{\Delta^{1 \to 2}}$ | $\mathbf{\Delta^{2 \to 3}}$ |
|---|---|---|---|
| $x_1$ (Age) | +2 | +3 | +2 |
| $x_2$ (Income) | +2 | +3 | +5 |
| $x_3$ (Credit Score) | −10 | −5 | −15 ($\times$) |
| $x_4$ (Number of Children) | 0 | +1 | 0 |

## 5 Materials and Methods

To evaluate the proposed DPG-da framework, we designed a comprehensive experimental protocol combining violation-prone, synthetic imbalanced, and benchmark datasets. Our methodology aims to test (i) data realism through adherence to domain constraints, (ii) classification performance under varying augmentation levels, and (iii) interpretability of synthetic samples. This section details the DPG-da implementation, baseline over-sampling algorithms and datasets.

### 5.1 DPG-da Implementation

The implementation of DPG-da is publicly available[1]. The codebase provides a reproducible workflow for generating realistic, constraint-compliant synthetic samples in imbalanced tabular datasets. To avoid data leakage, all augmentation procedures were applied exclusively to the training set; the surrogate model was trained only on training data, and augmented samples were added back to this set before classifier training. The test set remained untouched to ensure fair evaluation.

---

[1]Anonymized

The stages of the implementation are as follows:

1. **Surrogate Model Training:** A tree-based ensemble (RF) is trained to approximate the classification behaviour of the dataset, providing interpretable decision predicates.

2. **DPG Construction:** Decision predicates are extracted from the surrogate model to delineate class-specific feasible regions, supported by the DPG library [2].

3. **GA-Based Augmentation:** Synthetic samples are generated by evolving a population of candidate solutions through a GA.

4. **Fitness Function and Optimization:** The GA is guided by a composite fitness function promoting (i) strict adherence to DPG-derived constraints, (ii) diversity among synthetic samples, and (iv) sparsity of modifications relative to a query sample. In the current implementation, these objectives are combined via a weighted sum with $(w_1, w_2, w_3) = (2.0, 1.0, 3.0)$ (see Appendix B).

## 5.2 Baseline Algorithms

We compared it against 12 established over-sampling techniques: classical interpolation-based methods (SMOTE, SMOTE-ENN, SMOTE-SVM), advanced variants from the *smote-variants* library (DE, SMOTE-POLYNOM, MSMOTE, SMOTE-LVQ), generative approaches for tabular data (CTGAN, TVAE, COPULA-GAN) and ROS. Baselines were run with standard hyperparameters to ensure fair comparison. A summary is provided in Table 2 in the Appendix.

Recent advances in tabular generation also include diffusion-based models (e.g., TabDDPM (Kotelnikov et al., 2023)), which represent a promising direction for future work, though they are not included in our current benchmark.

## 5.3 Violation-Prone Datasets

To evaluate DPG-da under constraint-sensitive scenarios, we curated three types of violation-prone datasets:

**Synthetic violation-prone datasets:** We synthesized a set of datasets covering domains such as healthcare, finance, manufacturing, fraud detection, energy grid, and education. Features were sampled from plausible distributions, and domain-specific rules were encoded to evaluate violation-space adherence.

**Real-world imbalanced datasets:** Tabular datasets from the KEEL repository (Derrac et al., 2015) with imbalance ratios between 1.5 and 9 (e.g., Wisconsin and Pima) (Fernández et al., 2008). These allow testing on naturally occurring imbalances.

**Synthetic imbalanced datasets with noise and borderline examples:** From Napierała et al. (2010), including three minority-class shapes (*subclus*, *clover*, *paw*) to simulate challenging scenarios with noise and non-linear decision boundaries.

These datasets allow us to evaluate the over-sampling techniques ability to ensure generated data realism, by verifying that synthetic samples respect domain-specific constraints and remain semantically valid. A complete summary of the data and their valid feature ranges is provided in Appendix C.1.

## 5.4 Benchmarking Datasets

Furthermore, to evaluate the classification performance on augmented data from these over-samplers, we also selected 27 benchmark datasets from (Ding, 2011) and the *imbalanced-learn* library, spanning diverse domains, imbalance ratios, sample sizes, and feature dimensionalities. These datasets allow comprehensive assessment of method performance under realistic, challenging conditions. A summary is provided in Appendix C.2.

---

[2]https://github.com/LeonardoArrighi/DPG

# 6 Results and Discussions

Following the experimental protocol described in Section 5, we compare our method against established over-sampling baselines, spanning both classical interpolation-based techniques and recent generative models. The evaluation focuses on three complementary aspects: (i) the realism of augmented data, i.e., adherence to domain constraints; (ii) classification performance across multiple augmentation levels; and (iii) interpretability of the generated instances. This approach enables a holistic assessment of predictive performance alongside the plausibility and transparency of synthetic samples.

## 6.1 Ensuring generated data realism

We evaluate DPG-da alongside baseline over-sampling methods using three widely adopted classifiers: Logistic Regression, k-Nearest Neighbors (kNN), and Decision Trees. These classifiers represent a spectrum of model complexities and inductive biases: Logistic Regression assumes linear separability, kNN relies on local neighbourhoods and is sensitive to sample distribution, and Decision Trees capture non-linear relationships but can overfit to noisy or artificially augmented instances.

Each method is applied at three augmentation levels (15%, 30%, and 50% minority class proportion relative to the total dataset), and classifiers are trained on the resulting augmented datasets. To account for variability introduced by the random nature of over-sampling and model initialization, each experiment is repeated ten times per dataset and classifier combination.

To assess data realism, we check whether augmented samples adhere to the inherent constraints of each dataset. A violation is defined as any synthetic instance with feature values that are unrealistic or semantically inconsistent with the dataset domain. The number of violations serves as a quantitative measure of how well each method respects dataset constraints.

Figure 3 presents a heatmap of the normalized violation rates across over-sampling methods and representative datasets. SMOTE-LVQ exhibits the highest violation rates, with 0.497 in Wisconsin (98.6 violations), 0.253 in Pima (53.3), 0.045 in Fraud Detection (10.4), and roughly 0.005–0.041 in Quality Control, Finance, Clover, and Iris0. DE also generates violations, though generally at lower rates (e.g., 0.091 in Wisconsin, 0.129 in Pima, 0.109 in Fraud Detection, and smaller rates elsewhere). SMOTE-SVM shows moderate violations, particularly in Wisconsin (0.048) and Fraud Detection (0.036), while SMOTE-POLYNOM primarily affects Pima (0.192). All other datasets and methods exhibit negligible violation rates.

To illustrate these violations in detail, Appendix D.1 shows a pair-wise plot of the Wisconsin dataset augmented with SMOTE-LVQ. Several synthetic samples lie outside valid ranges, particularly in features such as *Mitoses*, *NormalNucleoli*, and *MarginalAdhesion*, highlighting the practical impact of constraint violations.

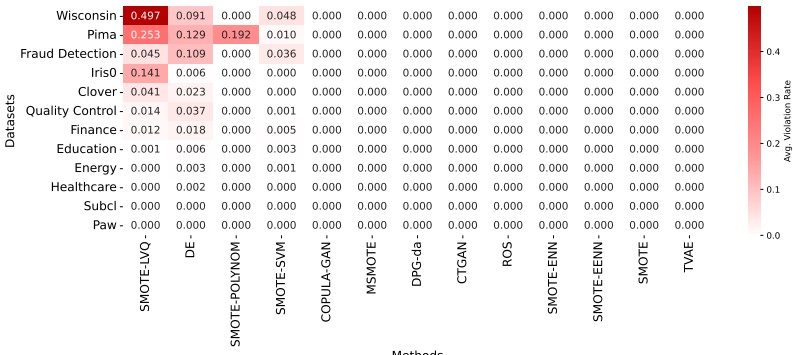

Figure 3: Heatmap of normalized constraint violation rates per over-sampling method and dataset. Violation rate is computed as the number of violations divided by the number of synthesized samples, averaged across repeated runs.

These findings demonstrate that some widely used over-sampling techniques can produce semantically invalid samples, especially in datasets with complex feature interdependencies. In contrast, DPG-da, which incorporates constraint-awareness into the augmentation process, avoids such violations, underscoring the importance of enforcing data realism when generating synthetic minority-class instances in sensitive domains.

## 6.2 Performance Analysis

### 6.2.1 Classification improvement

We assess classification performance using the macro F1-score and runtime, reporting the average across 10 repetitions. The F1-score is particularly suitable for imbalanced classification as it balances precision and recall, offering a more informative metric than accuracy when class distributions are skewed. For each dataset and augmentation level, results are first averaged over 10 independent runs; figures then report averages across the 27 datasets to provide a global comparison between methods. Dataset-level results are reported separately in Appendix F (Table 8).

To avoid any potential bias introduced by the surrogate model, data augmentation and downstream evaluation were strictly separated. The surrogate model used to generate synthetic samples was trained exclusively on a holdout portion of the training data (80%), while all downstream classifiers were evaluated on an untouched test set that was never exposed to the surrogate or the augmented data during training. This design ensures that increasing augmentation levels do not bias the evaluation toward surrogate-learned decision boundaries, and that observed performance changes reflect genuine generalization effects.

For completeness, we also evaluated the predictive quality of the surrogate model itself. The random forest used to extract decision predicates achieves an average macro F1-score of $0.72 \pm 0.16$ on the held-out test sets across the 27 datasets. We emphasize that this value is reported solely to assess surrogate fidelity and is not directly comparable to the 71% baseline shown in Figure 4, which corresponds to downstream classifiers trained on the original, non-augmented data. The surrogate is never used as a final predictor; rather, its role is limited to defining feasible regions for augmentation of the DPG-da method.

Across 27 imbalanced datasets, evaluated at three augmentation levels, Figure 4 summarizes the results. Most methods showed a gradual decline in F1-score as augmentation increased, indicating that over-sampling primarily introduced non-descriptive or noisy instances.

Among the evaluated methods, DPG-da, SMOTE-SVM, and TVAE consistently maintained high F1-scores across all augmentation levels, while also outperforming the baseline performance of 71%. This demonstrates their robustness to increasing amounts of synthetic data and highlights their suitability for imbalanced classification tasks under varying augmentation conditions. In contrast, MSMOTE exhibited moderate sensitivity to augmentation, while SMOTE and SMOTE-ENN showed notable performance degradation at higher sampling percentages.

To determine whether the observed differences in predictive performance were statistically significant, we conducted a Friedman test ($\alpha = 0.05$) followed by a Nemenyi post-hoc test (Demsar, 2006). For each dataset–method pair, the F1-scores used in the statistical analysis correspond to the average macro F1-score across all three augmentation levels (15%, 30%, and 50%), thereby providing a single aggregated performance measure per method and dataset. The Friedman test confirmed significant differences among the eight populations ($p < 0.001$), providing formal evidence that performance varies across augmentation methods. Appendix Table 6 reports each method's median (MD), median absolute deviation (MAD), and mean rank (MR).

We considered only methods that did not produce violations (described in Section 6.1). The critical difference (CD) diagram (Figure 5) shows the post-hoc analysis (CD = 1.167). DPG-da (MR = 2.531) achieves the highest overall rank and is statistically superior to all other methods. TVAE (MR = 3.704) and ROS (MR = 3.975) form a high-performing cluster with SMOTE (MR = 4.506), where differences within the group are not statistically significant. MSMOTE (MR = 5.494), SMOTE-ENN (MR = 5.494), CTGAN (MR = 5.272), and COPULA-GAN (MR = 5.025) form a lower tier, with no significant differences observed within this cluster. These results reinforce the earlier observation that DPG-da consistently outperforms competing

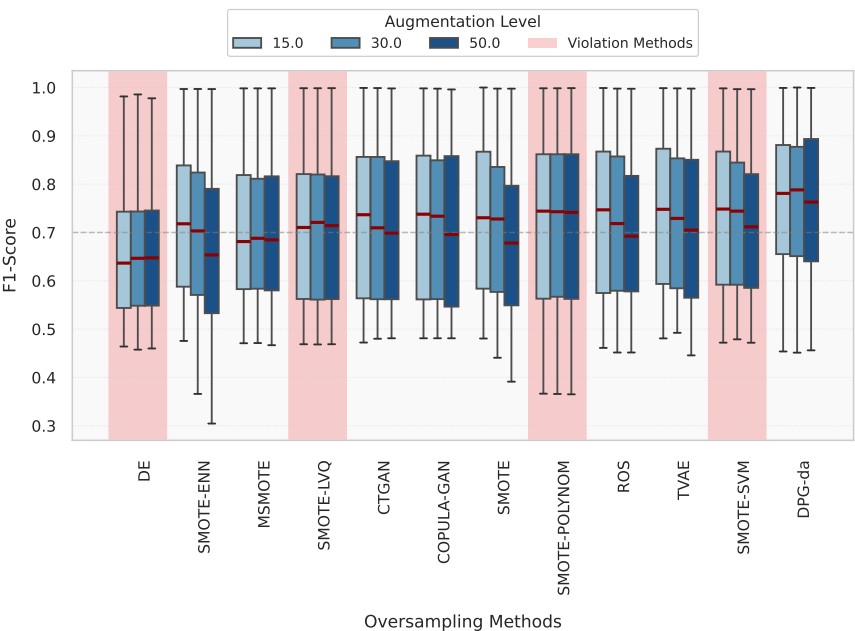

Figure 4: Mean classification performance across augmentation methods and sampling percentages. Red background areas indicate methods that violated constraints (DE, SMOTE-LVQ, SMOTE-POLYNOM, and SMOTE-SVM). The dashed line indicates the average performance of the classifiers on the original datasets without augmentation.

over-sampling techniques, demonstrating that constraint-aware augmentation produces both realistic and effective synthetic samples for imbalanced classification.

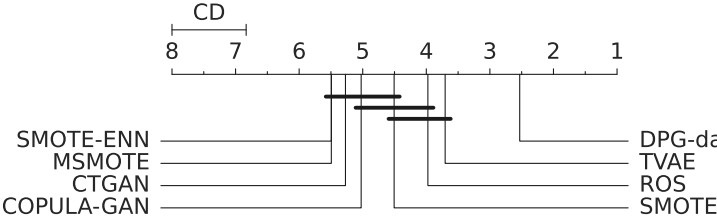

Figure 5: Critical difference diagram for F1-score performance. Methods not connected by a bar differ significantly (Nemenyi test, $\alpha = 0.05$, CD $= 1.167$).

### 6.2.2 Runtime

Figure 6 shows the log-scaled average runtime for generating synthetic samples across all evaluated over-sampling methods.

The interpolation-based methods, ROS, SMOTE, and SMOTE-POLYNOM, are extremely fast, with runtimes below 0.060 seconds. MSMOTE (0.072 seconds), SMOTE-ENN (0.740 seconds), and SMOTE-LVQ (2.520 seconds) require slightly more time due to additional operations such as noise filtering or SVM-based boundary computation.

Generative approaches, CTGAN (1.410 seconds), TVAE (1.120 seconds) and COPULA-GAN (5.240 seconds), introduce moderate overhead from model fitting and sample generation, but remain practical for typical

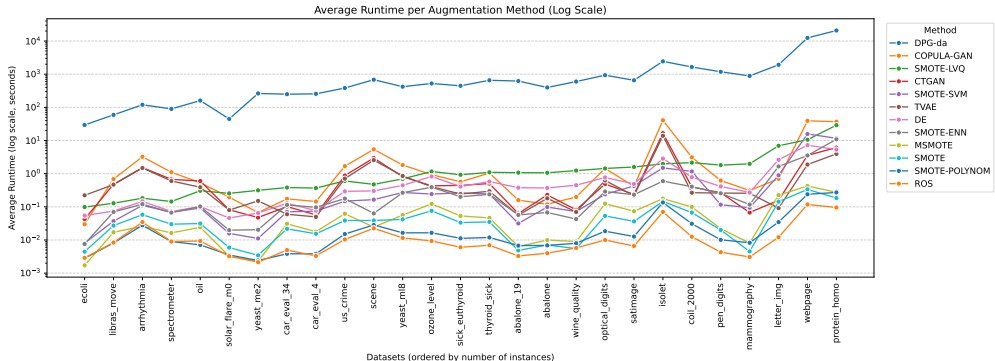

Figure 6: Log scaled Time performance of the augmentation methods.

tabular datasets. DE and SMOTE-SVM similarly incur small additional costs (0.950 and 1.250 seconds, respectively).

By contrast, DPG-da is substantially slower (1,809.45 seconds) due to its genetic algorithm optimization and the need to enforce DPG-derived constraints. This clearly illustrates the trade-off: DPG-da delivers highly realistic and interpretable synthetic data, but at the cost of considerably longer runtime, whereas all other methods remain extremely fast and scalable. We discuss the limitations of our proposal in the Appendix 7.

### 6.3 Interpreting augmentation

The final part of our analysis examines the interpretability of the proposed DPG-da augmentation method, which we define as the traceability and interpretation of the synthetic data generation process. Since DPG-da is grounded in the DPG framework, it inherits a transparent mechanism that reveals how new samples are produced.

To illustrate this property, we first present a heatmap (Figure 7) showing the average change ($\Delta$) in feature values between consecutive generations for the best individual in each generation of the evolutionary process on the Abalone dataset. Each cell reflects the magnitude of change for a specific feature, highlighting which attributes were most actively modified during the search.

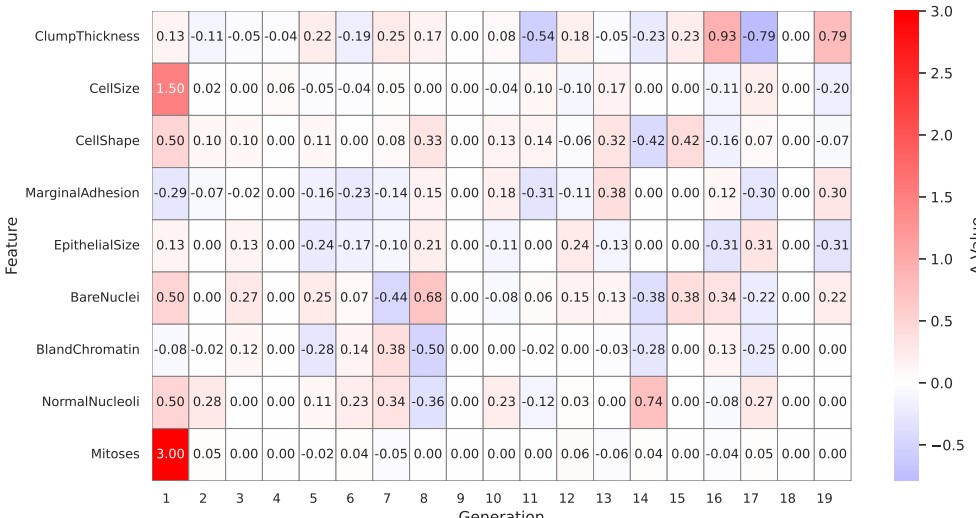

Figure 7: Heatmap of the evolutive process in the Wisconsin dataset. Showcasing the change ($\Delta$) in feature values between generations in the GA.

The query sample, represented by the vector:

$$\mathbf{x}^q = [5.00,\ 1.00,\ 1.00,\ 4.00,\ 2.00,\ 1.00,\ 3.00,\ 1.00,\ 1.00] \tag{2}$$

Was augmented into a synthetic sample:

$$\mathbf{x}^a = [5.98,\ 2.56,\ 2.58,\ 3.49,\ 1.65,\ 2.92,\ 2.30,\ 3.15,\ 4.05] \tag{3}$$

corresponding to the features: Clump Thickness, Cell Size, Cell Shape, Marginal Adhesion, Epithelial Size, Bare Nuclei, Bland Chromatin, Normal Nucleoli, and Mitoses.

All features were potentially adjusted by the GA, with the largest changes occurring in the most discriminative attributes for the target class. Clump Thickness, Cell Size, and Cell Shape are moderately modified to remain consistent with the DPG constraints. Continuous features such as Bare Nuclei, Normal Nucleoli, and Mitoses experience larger adjustments, reflecting targeted exploration of the feasible minority-class region. This pattern demonstrates how DPG-da selectively modified key features in this run while preserving the overall structure of the query sample, ensuring realistic and interpretable synthetic data.

We now turn to examples that trace how a single query sample is augmented into a synthetic counterpart. Figure 8 shows this process for three datasets: Pima, Ecoli, and Wisconsin. Each subplot visualizes the cumulative change applied to each feature across generations of the GA.

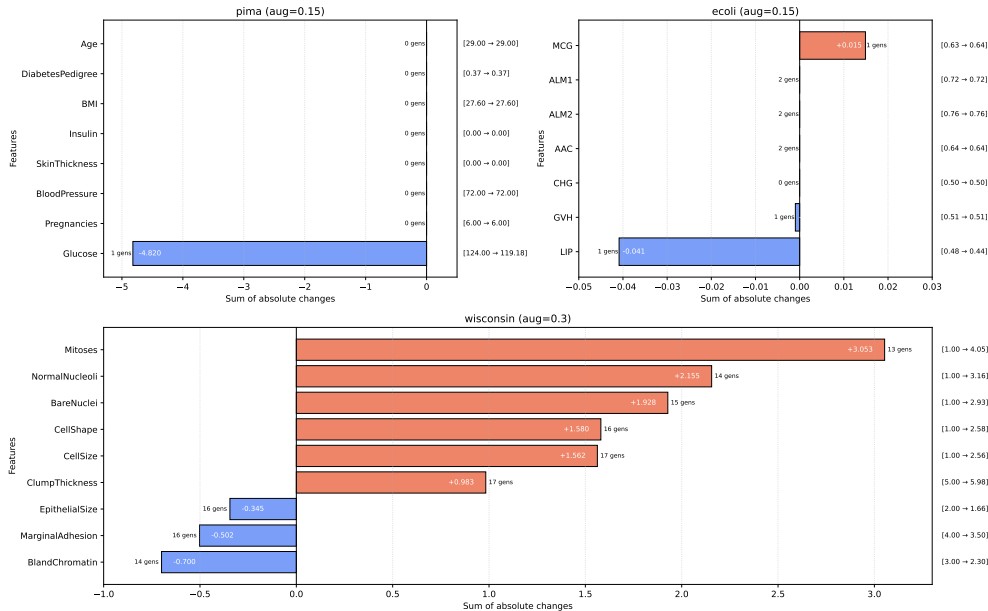

Figure 8: Interpretable feature changes during the genetic algorithm evolution. The top row shows Pima (left) and Ecoli (right), while the bottom row shows Wisconsin. Each horizontal bar represents the absolute difference of a feature from the first to the last generation. The white numbers inside the bars indicate the magnitude of the change, the black numbers outside the bars indicate the number of generations with non-zero change, and the text to the right shows the query-to-augmented transformation for each feature.

**Pima.** The Pima Indian Diabetes dataset describes patient medical measurements used for diagnosing diabetes. In our example, the query sample is already from the minority (diabetes) class, and the augmented instance remains almost identical, differing only in the *Glucose* feature. This adjustment is clinically plausible: blood glucose is one of the most discriminative variables for diabetes diagnosis, and a shift still yields a sample consistent with the minority class. Importantly, other attributes such as *Age* and (number of) *Pregnancies* remain unchanged, showing that DPG-da preserves the original patient profile while exploring variability within the minority region.

**Ecoli.** The Ecoli dataset concerns protein localization sites. Features such as *LIP* (signal peptide), *GVH* (Von Heijne signal sequence), and *MCG* (McGeoch method for signal sequence recognition) describe biochemical properties relevant to protein sorting. Here, the minority-class query is augmented with only small perturbations to *LIP* and *MCG*, while all other features remain unchanged. These modifications are subtle but biologically meaningful: since signal peptide properties are highly influential for localization, adjusting them slightly generates a new synthetic protein record that stays within the minority localization class but introduces plausible diversity.

**Wisconsin.** The Wisconsin Breast Cancer dataset encodes cytological measurements of cell nuclei obtained from fine needle aspirates. In this case, the minority-class query is augmented with broader changes. Features such as *Clump Thickness*, *Cell Size*, and *Cell Shape* are moderately adjusted, while continuous variables like *Bare Nuclei*, *Normal Nucleoli*, and *Mitoses* undergo larger shifts. These modifications are biologically interpretable: nuclei with irregular shapes, prominent nucleoli, and increased mitotic activity are characteristic of malignant tumors, the minority class in this dataset. The augmented sample thus represents a realistic new patient record that preserves minority-class pathology while exploring alternative manifestations of malignancy.

Together, these three examples highlight how DPG-da adapts its augmentation strategy to the context of each dataset, only small feature changes are introduced. When larger structural differences are needed (Wisconsin), the method applies broader but still interpretable modifications. In all cases, the augmentation remains transparent and traceable, producing synthetic data that not only looks realistic but also aligns with the underlying semantics of the domain.

## 7 Limitations

Although DPG-da demonstrates notable improvements over conventional over-sampling approaches, certain limitations should be acknowledged.

First, the method introduces a higher computational burden. The extraction of decision predicates from a surrogate model, followed by the use of a GA for optimization, increases runtime compared to classical interpolation-based methods. This effect is particularly evident in large-scale or high-dimensional datasets, which may limit the applicability of the method in time-constrained scenarios. While we used a vanilla GA to provide a straightforward understanding of our approach, this step could be replaced with a more lightweight optimization method to reduce computational costs.

Second, the effectiveness of the extracted constraints depends directly on the quality of the surrogate model. If the surrogate does not approximate the original decision boundaries with sufficient accuracy, the derived constraints may either be too restrictive or misrepresent the structure of the data. This dependence is particularly critical in problems with highly non-linear or complex decision boundaries, potentially limiting the validity of the generated samples.

Third, the procedure introduces an additional level of methodological complexity. The framework ensures that generated samples adhere to identified constraints, but the implementation requires more effort and expertise than simpler alternatives. This complexity is further compounded by the design of the fitness function used in the GA. While our current approach provides a straightforward mechanism to balance multiple objectives, it relies on manually chosen weights, which may affect the quality of the generated data. A more principled multi-objective optimization method (e.g., NSGA-II) could address this limitation, but would introduce additional complexity and computational cost.

Despite these limitations, several aspects of DPG-da mitigate their impact. The higher computational cost may be acceptable in application domains where the validity and consistency of augmented data are critical. Surrogate-derived constraints ensure that new samples remain coherent with the learned decision logic, reducing the risk of unrealistic or infeasible instances. Additionally, the structured nature of the augmentation process facilitates transparency and auditability, which are increasingly important in contexts requiring reliability and compliance. Overall, DPG-da represents a trade-off between computational efficiency and the assurance of constraint-adherent, consistent data augmentation.

## 8 Conclusion

We introduced DPG-da, a novel data augmentation method for imbalanced classification that integrates predictive performance with inherent interpretability. By leveraging decision pattern graphs to define rule-based boundaries, DPG-da generates instances that are semantically valid and transparent, avoiding the unrealistic samples often produced by traditional over-sampling techniques. Empirical results across benchmark and violation-prone datasets demonstrate that DPG-da consistently outperforms widely used over-sampling methods, achieving higher F1 scores. Future work may explore scalability to larger datasets, integration with deep learning pipelines, and incorporation of domain-specific constraints for high-stakes applications.

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

## A  Baseline Over-sampling Methods

The baseline over-sampling methods considered in this work, along with the proposed DPG-da, are summarized in Table 2, highlighting their main features and objectives.

Table 2: Summary of over-sampling methods compared in this study. Methods range from classical interpolation-based techniques to recent generative approaches.

| Method | Description |
| --- | --- |
| ROS | Random over-sampling; simple, replicates existing minority instances. |
| SMOTE | Interpolation-based over-sampling (Chawla et al., 2002); improves class balance. |
| SMOTE-ENN | SMOTE followed by Edited Nearest Neighbors (Batista et al., 2004); reduces noise. |
| SMOTE-SVM | Focuses augmentation near decision boundaries (Bunkhumpornpat et al., 2009). |
| MSMOTE | Targets difficult minority regions for improved discrimination (Hu et al., 2009). |
| SMOTE-POLYNOM | Captures non-linear feature relationships (Gazzah & Essoukri Ben Amara, 2008). |
| DE | Differential evolution-based over-sampling (Chen et al., 2010); enhances diversity. |
| SMOTE-LVQ | Selects informative minority prototypes for over-sampling (Nakamura et al., 2013). |
| CTGAN | GAN-based model for tabular data; learns conditional distributions (Xu et al., 2019). |
| TVAE | Variational Autoencoder for tabular data; handles mixed feature types (Xu et al., 2019). |
| COPULA-GAN | Combines copula modeling with GANs; balances flexibility and statistical fidelity (Xu et al., 2019). |
| **DPG-da** | Ensures domain constraint adherence, interpretable, avoids over-generalization. |

## B  Ablation Study

We conducted a systematic ablation study by evaluating all $3 \times 3 \times 3 = 27$ possible combinations of the three weighting factors from 1.0 to 3.0: distance ($w_1$), sparsity ($w_2$), and constraints ($w_3$). Figure 9 reports the top-performing configurations ranked by macro F1-score. While some configurations with higher values of $w_1$ (distance factor) consistently appear near the top, the overall differences among the ten best settings are relatively small. This indicates that performance is not overly sensitive to any single factor, and instead depends on a balanced interaction between distance, sparsity, and constraints.

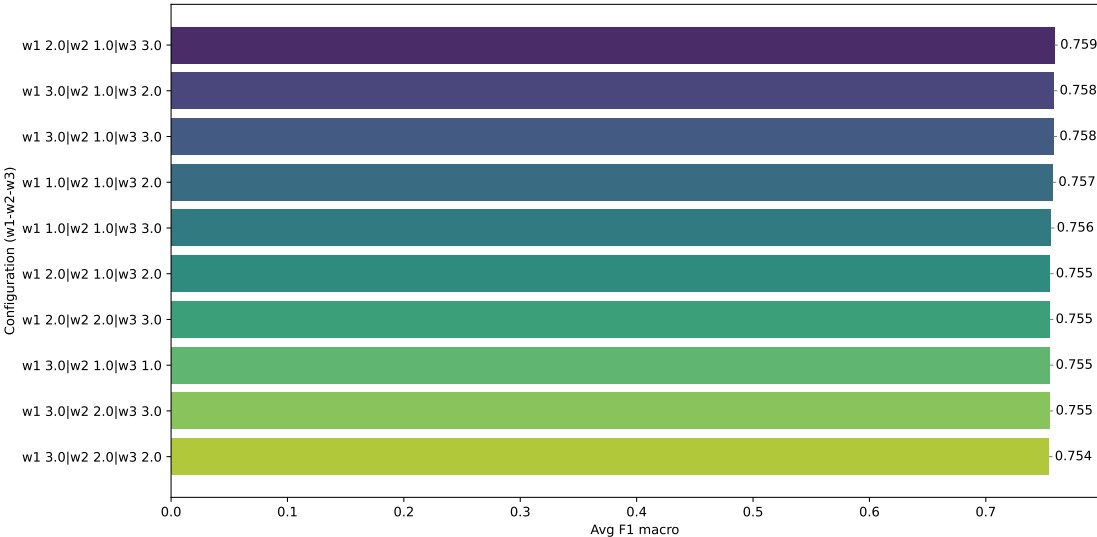

Figure 9: Top-performing ablation weight configurations for the optimization function, ranked by macro F1-score. Each configuration is denoted by its $(w1, w2, w3)$ tuple corresponding to distance, sparsity, and constraints factors. The results indicate which combinations yield the highest predictive performance across all datasets.

## C  Datasets

### C.1  Violation Benchmark

Table 3 provides an overview of the violation-prone datasets used in this study. Each dataset includes clearly defined feature ranges and domain-specific constraints, allowing systematic evaluation of whether synthetic samples generated by augmentation techniques respect valid and feasible values.

Table 3: Overview of datasets curated for the benchmark of violations

| Name | Ratio | #Samples | #Features |
|---|---|---|---|
| Healthcare | 3:1 | 1,000 | 4 |
| Finance | 1:1 | 1,000 | 4 |
| Quality Control | 2:1 | 1,000 | 4 |
| Fraud Detection | 3:1 | 1,000 | 4 |
| Energy | 1:1 | 1,000 | 3 |
| Education | 1:1 | 1,000 | 3 |
| Wisconsin | 2:1 | 683 | 9 |
| Pima | 2:1 | 768 | 8 |
| Iris0 | 2:1 | 150 | 4 |
| Subclus | 5:1 | 600 | 2 |
| Clover | 5:1 | 600 | 2 |
| Paw | 5:1 | 600 | 2 |

### C.2  Classification Benchmark

Table 4: Summary of the UCI imbalanced benchmarking datasets used in the experiments on Classifier performance. Ratio refers to the imbalance level.

| Name | Ratio | #Samples | #Features |
|---|---|---|---|
| ecoli | 8.6:1 | 336 | 7 |
| optical digits | 9.1:1 | 5,620 | 64 |
| satimage | 9.3:1 | 6,435 | 36 |
| pen digits | 9.4:1 | 10,992 | 16 |
| abalone | 9.7:1 | 4,177 | 8 |
| sick euthyroid | 9.8:1 | 3,162 | 25 |
| spectrometer | 11:1 | 531 | 93 |
| car eval_34 | 12:1 | 1,728 | 21 |
| isolet | 12:1 | 7,797 | 617 |
| us crime | 12:1 | 1,994 | 100 |
| yeast ml8 | 13:1 | 2,417 | 103 |
| scene | 13:1 | 2,407 | 294 |
| libras move | 14:1 | 360 | 90 |
| thyroid sick | 15:1 | 3,772 | 52 |
| coil 2000 | 16:1 | 9,822 | 85 |
| arrhythmia | 17:1 | 452 | 278 |
| solar flare m0 | 19:1 | 1,389 | 32 |
| oil | 22:1 | 937 | 49 |
| car eval 4 | 26:1 | 1,728 | 21 |
| wine quality | 26:1 | 4,898 | 11 |
| letter img | 26:1 | 20,000 | 16 |
| yeast me2 | 28:1 | 1,484 | 8 |
| webpage | 33:1 | 34,780 | 300 |
| ozone level | 34:1 | 2,536 | 72 |
| mammography | 42:1 | 11,183 | 6 |
| protein homo | 111:1 | 145,751 | 74 |
| abalone 19 | 130:1 | 4,177 | 10 |

# D    Constraint Extraction and Guided Augmentation

DPG-da automatically derives constraints from the classifier decision boundaries, which serve as the foundation for generating valid, semantically plausible synthetic samples. Constraints are represented as inequalities over the input features (e.g., bounds on feature combinations, threshold conditions) and are stored in a structured format (JSON) for programmatic evaluation.

As displayed in 5, for each dataset, we analyse the extracted constraints and report the values and types of constraints identified, illustrating how the feasible region for augmentation in DPG-da is defined. These constraints prevent generation of samples violating domain-specific or model-derived rules, ensuring that synthetic instances remain realistic while enriching minority-class coverage.

Table 5: Model-derived constraints extracted by DPG-da for the violation set of datasets.

| Dataset | Class | Feature Ranges / Constraints |
|---|---|---|
| Education | negative | Attendance: [50.68, 99.96]; Grades: [52.51, 99.56]; StudyHours: [0.47, 19.72] |
|  | positive | Attendance: [50.68, 99.96]; Grades: [52.51, 99.56]; StudyHours: [0.47, 19.72] |
| Energy | negative | Usage: [0.52, 2.47]; Baseline: [103.32, 495.35]; Voltage: [190.39, 249.85] |
|  | positive | Usage: [0.52, 2.44]; Baseline: [103.32, 498.23]; Voltage: [191.02, 249.85] |
| Finance | negative | CreditScore: [308.5, 845.0]; Income: [21340.5, 149246.5]; NumChildren: [0.5, 4.5] |
|  | positive | CreditScore: [309.0, 845.0]; Income: [21755.5, 149246.5]; NumChildren: [0.5, 4.5] |
| Fraud Detection | negative | TransactionAmount: [134.14, 9884.44]; TransactionTime: [0.89, 29.88] |
|  | positive | TransactionAmount: [134.14, 9955.39]; TransactionTime: [0.3, 29.88] |
| Healthcare | negative | BMI: [18.09, 39.81]; Cholesterol: [101.0, 296.0]; Age: [20.5, 88.5] |
|  | positive | BMI: [18.14, 39.81]; Cholesterol: [101.0, 294.0]; Age: [20.5, 88.5] |
| Quality Control | negative | Temperature: [80.25, 118.42]; Pressure: [5.17, 14.87]; Speed: [100.5, 298.5]; Vibration: [1.24, 9.96] |
|  | positive | Temperature: [80.25, 119.13]; Pressure: [5.17, 14.88]; Speed: [106.5, 298.5]; Vibration: [1.24, 9.96] |
| Wisconsin | negative | ClumpThickness: [3.5, 8.0]; CellSize: [2.5, 4.5]; CellShape: [1.5, 7.5]; MarginalAdhesion: [1.5, 7.5]; EpithelialSize: [1.5, 5.0]; BareNuclei: [1.5, 8.5]; BlandChromatin: [1.5, 4.5]; NormalNucleoli: [1.5, 8.5]; Mitoses: [4.0, 5.0] |
|  | positive | ClumpThickness: [3.5, 8.0]; CellSize: [2.0, 4.5]; CellShape: [2.5, 7.5]; MarginalAdhesion: [1.5, 7.5]; EpithelialSize: [2.5, 6.5]; BareNuclei: [1.5, 8.5]; BlandChromatin: [1.5, 4.0]; NormalNucleoli: [1.5, 7.5]; Mitoses: [1.5, 5.0] |
| Pima | negative | Preg: [0.5, 13.5]; Plas: [42.5, 180.5]; Pres: [25.0, 99.0]; Skin: [3.5, 41.0]; Insu: [34.0, 471.0]; Mass: [9.65, 47.55]; Pedi: [0.14, 1.31]; Age: [22.5, 66.5] |
|  | positive | Preg: [0.5, 13.5]; Plas: [42.5, 178.0]; Pres: [36.0, 93.0]; Skin: [6.0, 40.0]; Insu: [34.0, 235.0]; Mass: [9.65, 47.55]; Pedi: [0.14, 1.31]; Age: [22.5, 66.5] |
| Iris0 | negative | SepalLength: [4.9, 7.9]; SepalWidth: [2.0, 3.8]; PetalLength: [2.5, 3.5]; PetalWidth: [0.8, 1.8] |
|  | positive | SepalLength: [4.3, 5.8]; SepalWidth: [2.3, 4.4]; PetalLength: [1.7, 2.7]; PetalWidth: [-0.2, 0.8] |
| Subclus | negative | dim1: [52.5, 360.0]; dim2: [-118.0, 1312.5] |
|  | positive | dim1: [52.5, 360.0]; dim2: [-118.0, 1312.5] |
| Clover | negative | dim1: [-335.5, 377.5]; dim2: [-269.0, 402.0] |
|  | positive | dim1: [-335.5, 377.5]; dim2: [-367.5, 402.0] |
| Paw | negative | dim1: [139.5, 616.5]; dim2: [232.0, 854.0] |
|  | positive | dim1: [139.5, 616.5]; dim2: [232.0, 854.0] |

### D.1 Case Study: Violations

Figure 10 provides a detailed example of violations generated by an over-sampling method on the Wisconsin dataset. Many augmented samples fall outside the valid feature space, entering regions that are semantically invalid. This is particularly evident for features such as *Mitoses*, *NormalNucleoli*, and *MarginalAdhesion*, which take on negative values, an outcome that is unrealistic and inconsistent with domain knowledge.

Such violations are not only theoretically problematic but can also negatively impact downstream model performance. Classifiers trained on datasets containing semantically invalid samples may learn spurious patterns or overfit to unrealistic feature combinations, potentially reducing generalization to real-world data. This case study highlights the practical importance of constraint-aware augmentation methods like DPG-da, which systematically avoid these issues by ensuring that all generated samples remain within valid feature ranges.

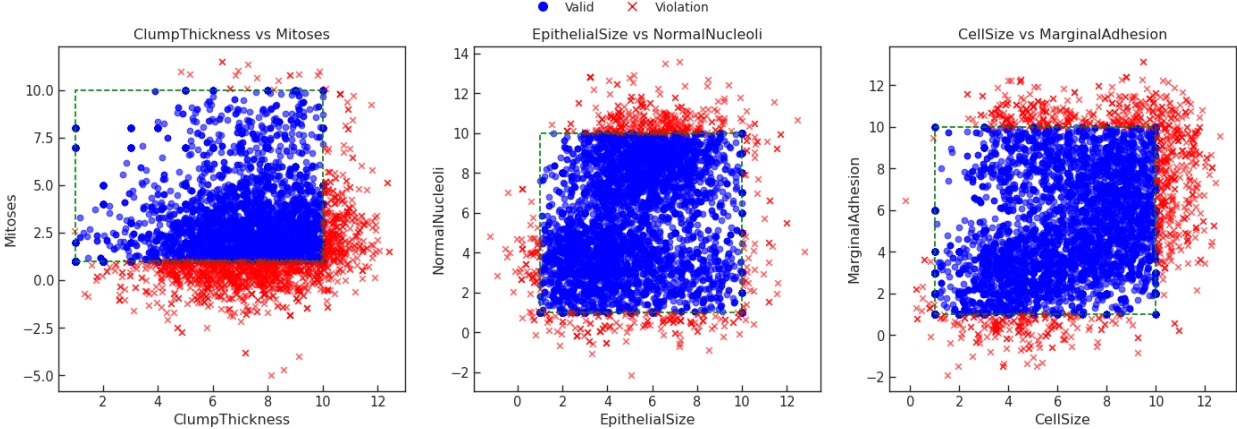

Figure 10: Pairwise scatter plots of SMOTE-LVQ augmented samples from the Wisconsin dataset. Blue circles indicate valid samples within feature constraints, red crosses indicate samples violating at least one feature range. Green rectangles show the valid region for each feature pair.

## E   Traceability across generations

To illustrate the traceability of our augmentation process, we track how candidate samples evolve during the GA. For each dataset, we record the feature-wise differences ($\Delta$) between successive generations relative to the initial query sample. This provides an interpretable view of how synthetic samples are formed: which features are modified, when changes occur, and whether evolution converges to stable solutions.

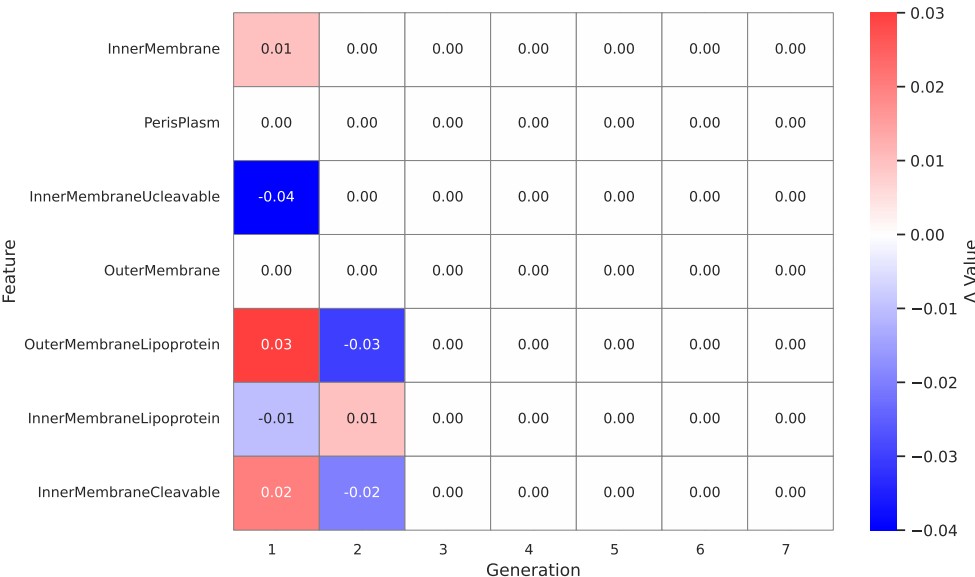

Figure 11: Heatmap of feature-wise changes (Δ) during the GA on the **Ecoli** dataset. Early generations explore a broad range of features, with substantial variation across most attributes. As evolution proceeds, the process converges, reflecting stabilization of the candidate samples within the feasible region.

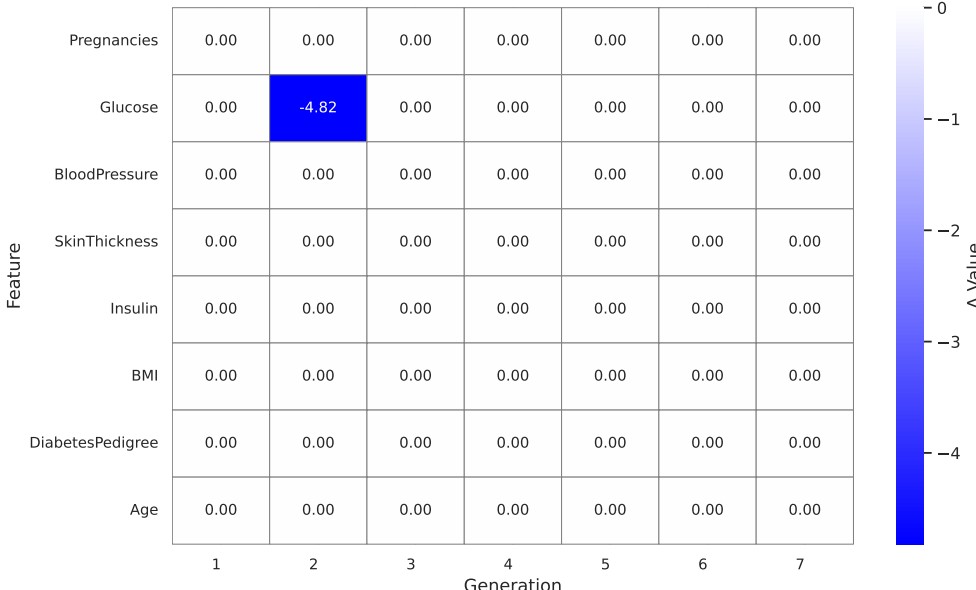

Figure 12: Heatmap of feature-wise changes (Δ) during the GA on the **Pima** dataset. Unlike Ecoli, the augmentation converges rapidly by modifying exclusively the *glucose* feature. This highlights a highly localized and interpretable transformation path, where minority samples are generated through a single, domain-relevant attribute shift.

## F   Performance Across Datasets

Below, in Table 6 is presented the detailed statistical results of the CD test.

Table 6: Detailed statistical results (MD, MAD, MR) complementing the CD diagram in Figure 5

| Method | MD ± MAD | MR |
| --- | --- | --- |
| DPG-da | $0.757 \pm 0.109$ | 2.531 |
| TVAE | $0.719 \pm 0.144$ | 3.704 |
| ROS | $0.728 \pm 0.131$ | 3.975 |
| SMOTE | $0.708 \pm 0.130$ | 4.506 |
| CTGAN | $0.722 \pm 0.135$ | 5.272 |
| COPULA-GAN | $0.691 \pm 0.158$ | 5.025 |
| SMOTE-ENN | $0.693 \pm 0.115$ | 5.494 |
| MSMOTE | $0.705 \pm 0.120$ | 5.494 |

To complement the aggregate performance analysis reported in Section 6, we provide a dataset-level comparison between baseline classification performance and performance obtained after augmentation with DPG-da. Figure 13 presents a plot where each point corresponds to one dataset, the x-axis denotes the baseline macro F1-score (no augmentation), and the y-axis denotes the macro F1-score achieved after applying DPG-da (averaged across augmentation levels and classifiers). The diagonal line indicates equal performance between baseline and augmented data.

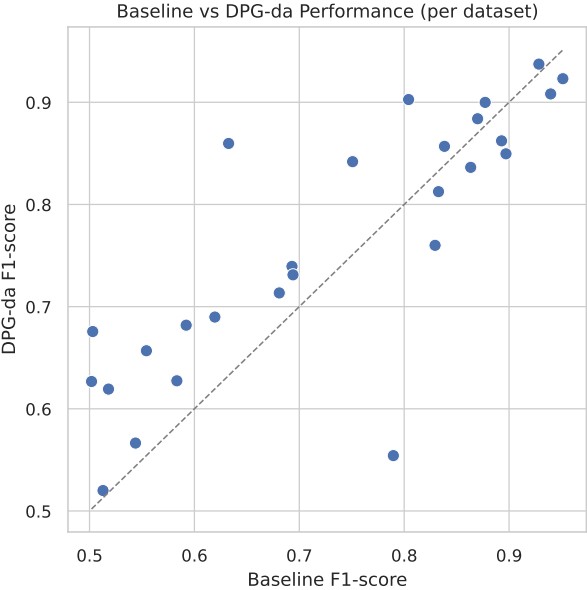

Figure 13: Baseline versus DPG-da macro F1-score comparison across datasets. Each point corresponds to one dataset, and the diagonal indicates equal performance.

Points above the diagonal correspond to datasets where DPG-da improves classification performance, while points below indicate performance degradation. The figure shows that DPG-da improves or preserves performance on the majority of datasets, with only a small number exhibiting minor decreases and a single dataset showing a marked drop, as displayed in Table 7.

This analysis confirms that the performance gains observed in the aggregate results are not driven by a small subset of datasets, but are instead consistent across most benchmark problems.

Table 7: Datasets for which the baseline (0% augmentation) achieves higher average F1-score than DPG-da.

| Dataset | Baseline F1 | DPG-da F1 |
|---|---|---|
| car_eval_34 | 0.893 | 0.862 |
| letter_img | 0.940 | 0.908 |
| mammography | 0.790 | 0.554 |
| pen_digits | 0.951 | 0.923 |
| protein_homo | 0.897 | 0.850 |
| sick_euthyroid | 0.829 | 0.760 |
| thyroid_sick | 0.833 | 0.813 |
| webpage | 0.863 | 0.836 |

To complement the F1-score analysis reported in Section 6, we provide a more detailed view of how precision and recall vary across augmentation methods and levels. Figure 14 presents boxplots of the precision and recall scores, averaged across datasets and classifiers, for each method at different augmentation percentages. The left panel shows precision, and the right panel shows recall. Red-shaded areas indicate methods that produced violations, highlighting the impact of constraint adherence on performance. Overall, the figure confirms that DPG-da consistently maintains a favourable balance between precision and recall, supporting the robustness of the F1-score improvements observed in the main results.

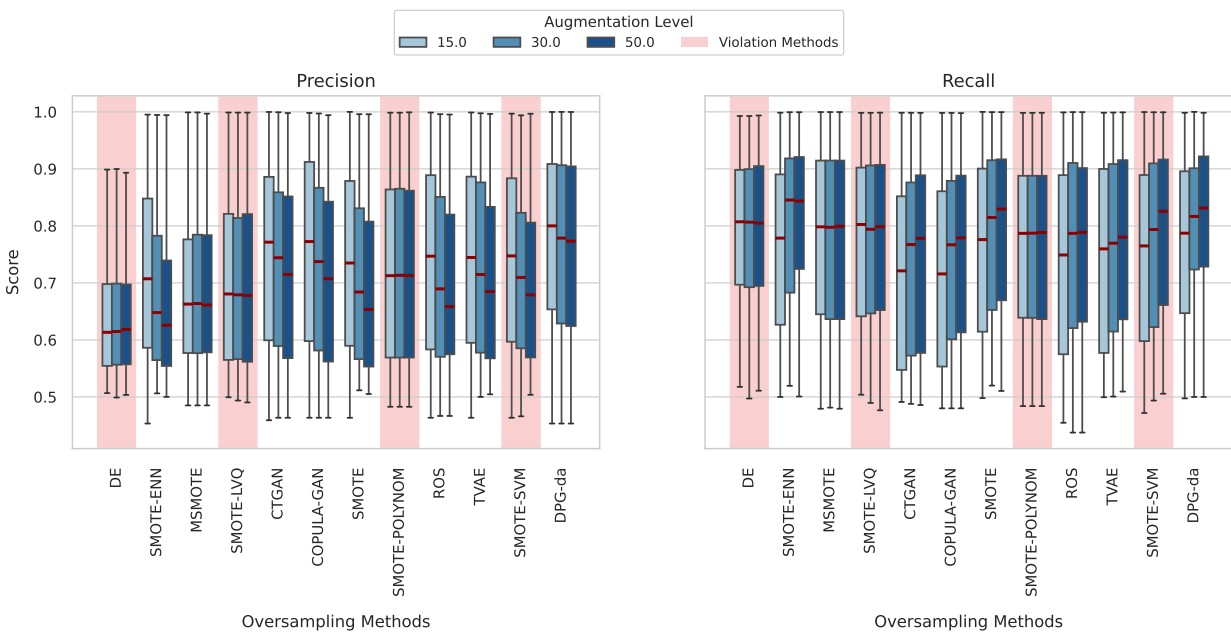

Figure 14: Distribution of precision (left) and recall (right) scores across augmentation methods and levels, averaged over datasets and classifiers. Red-shaded areas indicate methods that produced constraint violations. Both panels share a common y-axis labeled "Score" to facilitate comparison between metrics.

Table 8 reports the F1-Score for each dataset and over-sampling method, averaged over multiple augmentation levels. This table provides a detailed comparison of classifier performance under different augmentation strategies, highlighting both the effectiveness and limitations of each approach.

Table 8: F1-Score by dataset and over-sampling method, averaged over augmentation levels. The highest performance per dataset is in bold.

| Dataset | DE | SMOTE-ENN | MSMOTE | SMOTE-LVQ | COPULA-GAN | CTGAN | SMOTE-POLYNOM | SMOTE | ROS | TVAE | SMOTE-SVM | DPG-da |
|---|---|---|---|---|---|---|---|---|---|---|---|---|
| abalone | 0.600 | 0.602 | **0.614** | 0.584 | 0.588 | 0.599 | 0.601 | 0.594 | 0.593 | 0.598 | 0.608 | 0.566 |
| abalone_19 | 0.487 | 0.505 | 0.535 | 0.492 | 0.497 | 0.525 | 0.511 | 0.474 | 0.509 | 0.505 | 0.536 | **0.627** |
| arrhythmia | 0.639 | 0.659 | 0.590 | 0.617 | 0.629 | 0.672 | 0.651 | 0.716 | 0.617 | 0.654 | 0.663 | **0.860** |
| car_eval_34 | 0.821 | 0.829 | 0.823 | 0.866 | 0.874 | 0.889 | 0.872 | 0.870 | 0.881 | **0.921** | 0.882 | 0.862 |
| car_eval_4 | 0.853 | 0.866 | 0.897 | 0.885 | 0.902 | 0.900 | 0.895 | 0.917 | 0.896 | **0.934** | 0.907 | 0.903 |
| coil_2000 | 0.523 | 0.522 | 0.530 | 0.529 | 0.520 | 0.521 | 0.530 | 0.537 | 0.533 | 0.526 | 0.534 | **0.619** |
| ecoli | 0.656 | 0.762 | 0.733 | 0.755 | 0.781 | 0.751 | 0.760 | 0.751 | 0.722 | 0.782 | 0.749 | **0.842** |
| isolet | 0.729 | 0.810 | 0.826 | 0.862 | 0.872 | 0.876 | 0.841 | 0.870 | 0.861 | 0.868 | 0.849 | **0.884** |
| letter_img | 0.866 | 0.913 | 0.899 | 0.907 | 0.883 | 0.897 | 0.917 | **0.931** | 0.915 | 0.918 | 0.899 | 0.908 |
| libras_move | 0.667 | 0.826 | 0.833 | 0.805 | 0.831 | 0.862 | 0.855 | 0.834 | 0.840 | **0.884** | 0.867 | 0.857 |
| mammography | 0.683 | 0.740 | 0.759 | 0.752 | 0.721 | 0.745 | 0.750 | 0.732 | 0.772 | 0.762 | **0.794** | 0.554 |
| oil | 0.601 | 0.602 | 0.627 | 0.610 | 0.600 | 0.594 | 0.615 | 0.624 | 0.622 | 0.598 | 0.638 | **0.713** |
| optical_digits | 0.755 | 0.927 | 0.921 | 0.935 | 0.927 | 0.930 | 0.927 | 0.936 | 0.928 | 0.933 | 0.922 | **0.937** |
| ozone_level | 0.546 | 0.560 | 0.554 | 0.552 | 0.530 | 0.517 | 0.562 | 0.561 | 0.564 | 0.580 | 0.570 | **0.682** |
| pen_digits | 0.754 | 0.946 | 0.939 | 0.942 | 0.908 | 0.915 | 0.946 | 0.903 | **0.947** | 0.935 | 0.931 | 0.923 |
| protein_homo | 0.637 | 0.701 | 0.752 | 0.749 | 0.850 | **0.852** | 0.711 | 0.846 | 0.776 | 0.730 | 0.789 | 0.835 |
| satimage | 0.674 | 0.683 | 0.679 | 0.695 | 0.676 | 0.676 | 0.695 | 0.685 | 0.697 | 0.693 | 0.694 | **0.739** |
| scene | 0.572 | 0.538 | 0.568 | 0.573 | 0.553 | 0.559 | 0.561 | 0.588 | 0.580 | 0.574 | 0.589 | **0.657** |
| sick_euthyroid | 0.715 | 0.733 | 0.706 | 0.726 | 0.743 | 0.731 | 0.754 | 0.742 | 0.754 | 0.763 | **0.768** | 0.760 |
| solar_flare_m0 | 0.505 | 0.552 | 0.536 | 0.524 | **0.561** | 0.553 | 0.538 | 0.552 | 0.540 | 0.534 | 0.551 | 0.520 |
| spectrometer | 0.839 | 0.869 | 0.886 | 0.837 | 0.858 | 0.876 | 0.884 | 0.861 | 0.884 | 0.872 | 0.881 | **0.900** |
| thyroid_sick | 0.722 | 0.748 | 0.709 | 0.730 | 0.763 | 0.735 | 0.769 | 0.737 | 0.771 | 0.781 | 0.778 | **0.813** |
| us_crime | 0.678 | 0.681 | 0.696 | 0.702 | 0.704 | 0.689 | 0.697 | 0.701 | 0.699 | 0.697 | 0.720 | **0.731** |
| webpage | 0.731 | 0.692 | 0.731 | 0.808 | 0.781 | 0.780 | 0.746 | 0.807 | 0.783 | 0.753 | 0.769 | **0.836** |
| wine_quality | 0.558 | 0.582 | 0.587 | 0.563 | 0.601 | 0.577 | 0.591 | 0.583 | 0.606 | 0.580 | 0.615 | **0.627** |
| yeast_me2 | 0.553 | 0.597 | 0.609 | 0.591 | 0.583 | 0.524 | 0.602 | 0.567 | 0.602 | 0.612 | 0.634 | **0.690** |
| yeast_ml8 | 0.495 | 0.430 | 0.477 | 0.500 | 0.499 | 0.493 | 0.481 | 0.446 | 0.504 | 0.504 | 0.499 | **0.676** |

Below, Table 9 presents the average F1-score performance of the three classifiers (DecisionTree, LogisticRegression, and kNN) for the DPG-da augmentation method across all datasets and augmentation levels. Each row corresponds to a dataset, with the F1-scores reported sequentially for augmentation percentages of 0%, 15%, 30%, and 50%, respectively. The 0% columns correspond to the original (non-augmented) datasets. The RF column reports the macro F1-score of the random forest surrogate model used for constraint extraction, evaluated on the held-out test set, and is shown for reference only. RF is not used as a downstream classifier in the augmentation experiments. This table complements the analysis of F1-scores shown in the main figures by offering a detailed, per-classifier breakdown for DPG-da, allowing readers to clearly observe the performance gains or changes induced by data augmentation.

Table 9: DPG-da Macro F1-score for each dataset, classifiers, and augmentation percentage. Each row corresponds to one dataset, and columns are grouped by augmentation percentage (0%, 15%, 30%, 50%).

| Dataset | 0% | | | | 15% | | | 30% | | | 50% | | |
|---|---|---|---|---|---|---|---|---|---|---|---|---|---|
| | RF | DT | LR | kNN | DT | LR | kNN | DT | LR | kNN | DT | LR | kNN |
| abalone | 0.615 | 0.588 | 0.483 | 0.560 | 0.614 | 0.476 | 0.603 | 0.598 | 0.476 | 0.612 | 0.621 | 0.476 | 0.624 |
| abalone_19 | 0.497 | 0.510 | 0.498 | 0.498 | 0.634 | 0.564 | 0.748 | 0.640 | 0.505 | 0.642 | 0.648 | 0.502 | 0.757 |
| arrhythmia | 0.573 | 0.793 | 0.620 | 0.485 | 0.859 | 0.881 | 0.798 | 0.834 | 0.881 | 0.801 | 0.859 | 0.892 | 0.934 |
| car_eval_34 | 0.810 | 0.957 | 0.956 | 0.765 | 0.915 | 0.952 | 0.685 | 0.923 | 0.960 | 0.668 | 0.923 | 0.948 | 0.787 |
| car_eval_4 | 0.827 | 0.988 | 0.922 | 0.503 | 0.974 | 0.895 | 0.828 | 1.000 | 0.895 | 0.818 | 0.974 | 0.922 | 0.818 |
| coil_2000 | 0.537 | 0.555 | 0.492 | 0.507 | 0.585 | 0.509 | 0.672 | 0.612 | 0.554 | 0.728 | 0.634 | 0.534 | 0.745 |
| ecoli | 0.710 | 0.773 | 0.715 | 0.765 | 0.828 | 0.676 | 0.898 | 0.898 | 0.811 | 0.898 | 0.898 | 0.765 | 0.905 |
| isolet | 0.845 | 0.785 | 0.908 | 0.917 | 0.805 | 0.916 | 0.933 | 0.778 | 0.922 | 0.946 | 0.787 | 0.917 | 0.950 |
| letter_img | 0.969 | 0.967 | 0.866 | 0.986 | 0.963 | 0.829 | 0.988 | 0.956 | 0.790 | 0.981 | 0.948 | 0.742 | 0.976 |
| libras_move | 0.911 | 0.747 | 0.886 | 0.882 | 0.777 | 0.856 | 0.912 | 0.818 | 0.856 | 0.912 | 0.758 | 0.912 | 0.912 |
| mammography | 0.815 | 0.793 | 0.757 | 0.819 | 0.461 | 0.791 | 0.454 | 0.451 | 0.756 | 0.458 | 0.456 | 0.703 | 0.457 |
| oil | 0.623 | 0.666 | 0.746 | 0.631 | 0.781 | 0.669 | 0.718 | 0.726 | 0.662 | 0.739 | 0.735 | 0.640 | 0.751 |
| optical_digits | 0.922 | 0.893 | 0.912 | 0.980 | 0.908 | 0.918 | 0.992 | 0.911 | 0.918 | 0.990 | 0.893 | 0.919 | 0.987 |
| ozone_level | 0.532 | 0.582 | 0.614 | 0.580 | 0.731 | 0.581 | 0.675 | 0.705 | 0.660 | 0.730 | 0.672 | 0.638 | 0.744 |
| pen_digits | 0.979 | 0.972 | 0.885 | 0.997 | 0.977 | 0.832 | 0.999 | 0.970 | 0.798 | 0.999 | 0.969 | 0.765 | 0.999 |
| protein_homo | 0.888 | 0.868 | 0.910 | 0.913 | 0.850 | 0.914 | 0.784 | 0.850 | 0.910 | 0.784 | 0.859 | 0.911 | 0.783 |
| satimage | 0.761 | 0.754 | 0.492 | 0.834 | 0.786 | 0.515 | 0.840 | 0.808 | 0.585 | 0.855 | 0.821 | 0.560 | 0.884 |
| scene | 0.520 | 0.560 | 0.564 | 0.539 | 0.621 | 0.597 | 0.675 | 0.651 | 0.553 | 0.794 | 0.640 | 0.594 | 0.787 |
| sick_euthyroid | 0.884 | 0.897 | 0.823 | 0.768 | 0.905 | 0.804 | 0.605 | 0.877 | 0.791 | 0.604 | 0.878 | 0.763 | 0.611 |
| solar_flare_m0 | 0.561 | 0.509 | 0.544 | 0.486 | 0.512 | 0.486 | 0.574 | 0.499 | 0.526 | 0.558 | 0.512 | 0.523 | 0.488 |
| spectrometer | 0.858 | 0.828 | 0.924 | 0.880 | 0.899 | 0.922 | 0.874 | 0.858 | 0.943 | 0.874 | 0.911 | 0.943 | 0.874 |
| thyroid_sick | 0.908 | 0.935 | 0.816 | 0.747 | 0.914 | 0.801 | 0.706 | 0.943 | 0.824 | 0.710 | 0.932 | 0.820 | 0.664 |
| us_crime | 0.646 | 0.679 | 0.741 | 0.662 | 0.711 | 0.746 | 0.704 | 0.728 | 0.758 | 0.712 | 0.758 | 0.735 | 0.729 |
| webpage | 0.795 | 0.834 | 0.878 | 0.878 | 0.852 | 0.854 | 0.799 | 0.846 | 0.863 | 0.803 | 0.847 | 0.857 | 0.806 |
| wine_quality | 0.587 | 0.656 | 0.534 | 0.560 | 0.675 | 0.586 | 0.655 | 0.692 | 0.592 | 0.644 | 0.668 | 0.553 | 0.581 |
| yeast_me2 | 0.489 | 0.646 | 0.588 | 0.624 | 0.656 | 0.638 | 0.771 | 0.604 | 0.656 | 0.788 | 0.697 | 0.635 | 0.763 |
| yeast_ml8 | 0.533 | 0.502 | 0.495 | 0.512 | 0.699 | 0.499 | 0.730 | 0.684 | 0.596 | 0.769 | 0.721 | 0.640 | 0.742 |

## G    Performance Correlation between Dataset Properties

To further characterize the strengths and limitations of the augmentation methods, we analysed the three top-performing methods according to the experiments made in Section 4 (DPG-da, SMOTE-SVM and TVAE). We conducted a Spearman correlation analysis between dataset characteristics, namely number of features, number of instances, and imbalance ratio, and both classification performance and runtime (Figure 15).

The analysis reveals that SMOTE-SVM exhibits a slight negative correlation between the number of features and classification performance, which is consistent with the known limitations of SVM-based oversampling in high-dimensional spaces, where kernel boundaries become harder to define and synthetic samples may increase class overlap. In contrast, DPG-da shows a moderate positive correlation with the number of features, suggesting that its graph-based constraint extraction benefits from richer feature spaces, where predicate combinations provide more informative boundaries. TVAE, instead, remains largely neutral with

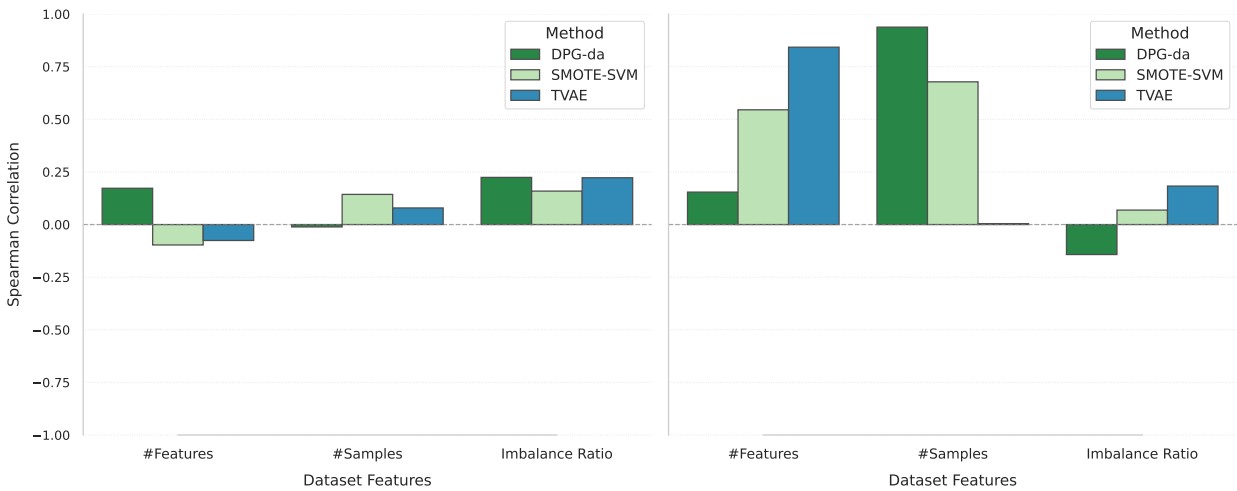

Figure 15: Spearman correlation between dataset characteristics (*Number of Features*, *Number of Instances*, and *Imbalance Ratio*) and two aspects: **left** the **F1-score performance** of the augmentation methods and **right** their **runtime**.

respect to feature dimensionality, reflecting the ability of generative models to capture joint distributions without relying on explicit feature-threshold splits.

When considering the number of instances, correlations with performance are generally weak: SMOTE-SVM and TVAE show a small positive trend, indicating that larger sample sizes provide more support for boundary construction or distribution learning, while DPG-da remains nearly unaffected, as its optimization depends more on predicate diversity than on sample count. Finally, with respect to the imbalance ratio, both DPG-da and TVAE display moderate positive correlations, highlighting their ability to take advantage of datasets with stronger class imbalance, either by exploiting DPG-derived constraints (DPG-da) or by better fitting minority distributions (TVAE). SMOTE-SVM, however, shows little sensitivity to imbalance ratio, reflecting its reliance on support vectors that may not fully capture minority class structure when imbalance is extreme.

With respect to runtime (Figure 15, right), clear method-specific trends can be observed. DPG-da exhibits a strong positive correlation with the number of instances and a more limited correlation with the number of features, reflecting the two main components of its computational cost: (i) surrogate model fitting and predicate extraction, whose cost increases with the size of the training data and the resulting complexity of the induced random forest, and (ii) the evolutionary optimization process, whose dominant cost scales with the number of fitness evaluations and the cost of evaluating predicate violations over an increasing number of samples. As dataset size grows, deeper trees and a larger set of predicates are typically induced, leading to increased runtime.

SMOTE-SVM shows positive correlations with both the number of features and the number of instances. This behaviour is consistent with the cost structure of SVM-based oversampling, where both training the SVM and computing distances in feature space become more expensive as dimensionality and dataset size increase, even though the synthetic sample generation itself remains relatively lightweight.

TVAE, in contrast, displays a clear positive correlation with the number of features, while being largely insensitive to the number of instances. This highlights the computational demands of training deep generative models in high-dimensional spaces, where increasing feature dimensionality leads to larger network architectures and longer convergence times, whereas moderate changes in dataset size have a smaller impact on training cost. For all methods, runtime shows limited sensitivity to the imbalance ratio, as class skew primarily affects data composition rather than algorithmic complexity.

Overall, these results explain why DPG-da achieves higher runtime gains over baselines on smaller or moderately sized datasets, while its computational cost increases more noticeably with dataset scale and surrogate model complexity. This trade-off highlights that DPG-da prioritizes constraint quality and augmentation fidelity over raw scalability, whereas SMOTE-SVM and TVAE exhibit more predictable scaling patterns driven by feature-space operations and neural network training dynamics, respectively.

