# OpenReview forum: "Close to Reality: Interpretable and Feasible Data Augmentation for Imbalanced Learning"
_TMLR — Rejected by TMLR_

### Review · Reviewer_UxQS · 2025-10-14

**Summary Of Contributions:**

The authors propose using a novel data augmentation strategy for imbalanced data that uses Decision Predicative Graphs to define class-specific feasible regions and then samples additional minority class data from the appropriate region using a genetic algorithm. Results on a wide range of example datasets suggest DPG-da performs well and is at least somewhat interpretable, but comes with substantial computational cost.

**Audience:**

Yes

**Audience Explanation:**

Class imbalance is a widespread problem across many applied disciplines.

**Claims And Evidence:**

Yes

**Claims Explanation:**

WHile I think there are interesting ideas here the methodology needs more detail/precision. Pseudocode might be helpful. Questions:
- how does DPG-da give a binary outcome/feasible region, rather than a probability, given that is based on a random forest?
- there seems to be some circularity in the sense that the random forest is trained on the (imbalanced) data, and then used to guide data augmentation. if the random forest overfits (or underfits), won't this bleed into the final model via the data augmentation?
- tabular data will often contain discrete/integer data: presumably this "constraint" will not be respected?
- why is the GA initialized uniformly at random rather than at x_q? (the latter would presumably help enable sparse augmentation)
- given the feasible region could be highly complex, how is it sampled from?
- "evaluates whether the candidate sample xa and the query sample xq are assigned to the same target class" is this the true target class for xq or the prediction?
- "Distance from the query sample": why is this just from xq? wouldn't we want xa that are different from all x in the minority class?

The spread (y axis) on Fig 4 is very large. Presumably much of this is due to variation across datasets (which isn't represented in the baseline). At least in supplement it would be useful to show F1 for baseline vs F1 for DPG-da as a scatterplot to see how often augmentation actually reduces performance. AUPR should also be reported since F1 is sensitive to small changes in the predicted mean.

**Requested Changes:**

Clarify the methods description to address the questions above.

Minor: some of the equations have additional space above them.

---

> ### Author Response · Authors · 2026-01-06
> **Response**
>
> First of all, we would like to thank the reviewer for their thoughtful feedback and positive assessment of the contribution. We have revised the manuscript to clarify the methodology and address the questions raised.
>
> DPG-da does not rely on probabilistic outputs of the random forest. Although a random forest can return class probabilities, our method uses its discrete predictions and, crucially, the decision predicates along tree paths. These predicates are converted into constraints defining a class-specific feasible region. Candidate samples are evaluated in a binary manner: they are feasible if they satisfy the extracted predicates and are class-consistent under the surrogate; otherwise, they are infeasible.
>
> Regarding potential circularity, the random forest is not used as the final predictor nor are its probabilities transferred to downstream classifiers. Its role is limited to defining predicate-based feasibility constraints. Sample generation is guided by a multi-objective fitness function that balances feasibility with locality (distance to the query point), sparsity, and diversity, reducing the risk of reproducing surrogate artifacts. Importantly, the surrogate is trained on a holdout portion of the training data, while downstream classifiers are evaluated on an untouched test set, mitigating bias toward surrogate-induced decision boundaries.
>
> We agree that severe surrogate misfit could influence the feasible region; however, feasibility is only one weighted component of the fitness function, and its influence can be moderated relative to other objectives. We now clarify this explicitly.
>
> The current implementation does not explicitly enforce discrete or integer feature constraints; however, this is an orthogonal extension that can be addressed through type-aware mutation operators without altering the framework. We acknowledge this limitation.
>
> The GA population is initialized uniformly within the feasible region to promote diversity and avoid premature convergence. Initializing at the query point alone risks restricting exploration. Sparsity and locality are still encouraged through the fitness function.
>
> We do not explicitly sample the feasible region; instead, we treat sampling as a search problem. Candidate points are iteratively proposed, evaluated for feasibility via predicate checks, and guided by fitness objectives, allowing exploration of complex or non-convex regions without explicit enumeration.
>
> Class consistency is determined by checking whether the surrogate’s prediction for a candidate matches the true target class of the query sample; we have clarified this in the text. While measuring distance to all minority samples could increase diversity, it would substantially increase computational cost and risk propagating noise from synthesized samples. Empirically, anchoring generation to individual query points already yields strong performance.
>
> In response to the reviewer’s suggestion, we added a scatterplot comparing baseline versus DPG-da F1-scores per dataset, along with complementary precision and recall boxplot to the Appendix of the document. While AUPR is informative, we focus on macro F1, precision, and recall for consistency with prior oversampling literature and because our contribution targets data generation rather than threshold calibration. AUPR-based evaluation is left for future work.
>
> We believe these revisions substantially improve clarity and address the reviewer’s concerns.

---

### Review · Reviewer_FDua · 2025-12-15

**Summary Of Contributions:**

This work aims to address the issue of ensuring constraint-enforced samples generations in over-sampling mechanisms. Where some of the existing works offer the interpretability feature, they lack domain constraints imposition or vice versa. To achieve both interpretability and domain-specific constraints imposition, they propose DPG-da, a constraint-aware augmentation method guided by Decision Predicate Graphs (DPGs). They show that the proposed interpretable and constraint-enforcing method yields better samples and as a result better classification performance for the imbalanced datasets.

**Audience:**

Yes

**Audience Explanation:**

Over-sampling is extensively used method for imbalanced datasets in ML, and the proposed method can advance the domain by making it more interpretable and constraint-adhering.

**Claims And Evidence:**

Yes

**Claims Explanation:**

Yes, the provided results show the interpretability of the features/samples, better abidance of the constraints and better classification performance. The baseline methods are also extensive (>10).

**Requested Changes:**

For Figure 4, it would be helpful to see how large the gain is over existing methods, same for the statistical significance results in Figure 5.

Also I wonder why the scores are averaged over 10 datasets, and not reported separately. would be interesting to see if gain is consistent across all datasets?

Given the runtime gain is higher than the other baselines, it is important to discuss how/if the runtime scales with the decision trees in random forest, or feature size etc?

---

> ### Author Response · Authors · 2026-01-06
> **Response**
>
> We thank the reviewer for the positive assessment of the contribution and for the constructive suggestions.
>
> To better illustrate the performance gains, we have expanded the appendix (Appendix G) with additional material. This includes (i) a scatterplot comparing baseline versus DPG-da performance on a per-dataset basis and (ii) a detailed table reporting per-classifier F1-scores for DPG-da across all datasets and augmentation levels. These additions complement Figures 4 and 5 by making both absolute improvements and dataset-level behavior explicit.
>
> We clarify that scores are not averaged over 10 datasets. Instead, for each dataset and augmentation level, results are averaged over 10 independent runs to account for stochasticity in both over-sampling and model training. Performance is then summarized across the 27 datasets to provide a global comparison, following standard multi-dataset benchmarking practice. Importantly, dataset-level results are fully reported in Appendix G (Table 6), allowing readers to verify that the observed gains are consistent across datasets rather than driven by a small subset. We have revised the manuscript to make this evaluation protocol explicit.
>
> Regarding runtime, we now include (Appendix H), a dedicated discussion of scalability. The computational cost of DPG-da decomposes into (i) surrogate training and predicate extraction, which scale with the number of trees and nodes in the random forest, and (ii) the evolutionary search, whose dominant cost scales with the number of fitness evaluations and the complexity of constraint checks, which in turn depend on feature dimensionality and constraint-set size. We additionally report empirical observations from our experiments showing how runtime correlates with these factors.
>
> We believe these revisions address the reviewer’s concerns and we would like to thank you again for your contribution.

---

### Review · Reviewer_CePR · 2025-12-23

**Summary Of Contributions:**

The authors consider the task of generating synthetic data samples, with the specific goal of compensating for class imbalance in tabular data sets. They highlight that existing methods for this task do not necessarily produce feasible/valid and interpretable datapoints. In this work, feasibility/validity is defined as the satisfaction of class-specific constraints; the constraints considered in this work are halfspaces in feature/data space. They formulate a novel method called Decision Predicate Graphs for Data Augmentation (DPG-da). In this method,

1. A tree-based surrogate model is trained on the available data/labels. This work focuses on random forest models.

2. The surrogate model is used to define class constraints. The authors formalize these class constraints in terms of decision predicate graphs (DPG’s). In this work, e.g. Figure 1, a DPG appears to be a single decision tree expressing the classification rules learned by the surrogate model, with the additional information of weights along edges indicating the frequencies of different occurrences in the training set. The authors use the decision boundaries produced by the surrogate model to define class-specific constraints, which later synthetic data must satisfy.

3. A population of synthetic data samples is generated by optimizing a composite objective using genetic algorithms. For every synthetic example $x_a$ which the user wishes to generate, the user must select a fixed, pre-specified existing training set sample $x_q$, which is used to define the optimization objective. The composite objective contains four terms, which ensure that (a) the synthetic sample $x_a$ satisfies the same class label, as predicted by the surrogate model,  as $x_q$; (b) the synthetic example satisfies the corresponding class constraints, as extracted from the DPG; (c) a diversity-promoting term which seeks to maximize the Euclidean distance between the synthetic example and $x_q$; (d) a sparsity-promoting term which seeks to make the difference vector $x_a-x_q$ sparse, for $x_a$ the synthetic example.

The authors compare DPG-da to existing over-sampling techniques. They consider benchmark datasets and generate augmented, synthetic data samples with greater degrees of balanced classes using DPG-da and other baseline techniques. They claim that when downstream classification methods are trained on these augmented datasets, DPG-da leads to the greatest improvement of the downstream classifier. They also validate empirically that, by construction, DPG-da does not generate any synthetic examples that violate the surrogate-model-defined class constraints.

Strengths:
- Class imbalance is a common challenge in applications of statistical and machine learning methods. The authors’ proposed method can be used as a stand-alone pre-processing step which can be easily incorporated into most classification workflows, making it a very broadly applicable method.

Weaknesses:
- There are core pieces of empirical comparisons absent in the current manuscript. In particular, the authors never validate that DPG-da improves performance over the surrogate model used to generate the augmented samples. This is a core question because, as discussed below, in the large-sample limit, one might expect downstream classifiers to be strongly biased towards the surrogate model’s decision boundaries.
- The data samples generated by this method will emulate the class decision boundaries produced by the surrogate model. Thus, any errors in the surrogate model are inherited by the downstream augmented dataset. Moreover, tree-based surrogate models will produce decision boundaries with very specific geometric properties (i.e., in this paper, coordinate-aligned hyperplanes in data space). For many natural data distributions, this will be a poor fit to the underlying data distribution, and thus generating a large volume of synthetic samples to “fill in” the regions of space defined by the decision tree will produce a data distribution that is very different from the ground truth. It is not clear that training on such an unrepresentative augmented dataset would improve downstream performance.

Core questions:
- Do output models trained with synthetic data have better classification performance than the surrogate model?
- Naively, as the number of augmented samples increases, one might hypothesize that the downstream trained optimizers are increasingly biased towards the boundaries learned by the surrogate model. Did the authors empirically probe any such behavior?
- The set of decision boundaries considered for formulating the constraints, namely, thresholding on the value of a single feature at a time, seems like a very strong constraint. Some immediate generalizations would include considering hyperplanes (i.e,. thresholding on linear combinations of features) or considering hyperplanes in an embedded feature space. Would such extensions be straightforward to incorporate into the DPG-da framework, or does the optimization routine rely heavily on the specific forms of the constraints used in this work?
- The constraint sets presented in Table 5 are all convex, connected sets, whereas in principle, random forests can yield class decision rules that are unions of disjoint regions, e.g.
$$\\{x_1 \leq 3, x_2 \geq 5\\} \vee \\{ x_1 \geq 10, x_2 < 5\\}$$
- Do such boundaries simply never arise in the datasets evaluated, or does the process of extracting DPG’s involve a step which causes all constraint regions to be connected?

Other questions:
- Which F1-scores are used in computing the critical difference results depicted in Figure 5? Are they F1 scores produced at all augmentation levels depicted in Figure 4 (e.g. 15, 30, 50), or only F1 scores for a specific augmentation level?
- For many tabular datasets, datapoints are implicitly constrained to have only integer values. It is my impression that DPG-da would not necessarily generate samples that satisfy such constraints, even when present in the training data. Is this correct? Are there reasonable modifications that would enable this extension?
- Is it necessary to use a random forest for the surrogate model? My impression is that the main motivations for this choice are the interpretability of the decision rules and the ability to efficiently evaluate distances to decision boundaries. Given some alternative classifier satisfying these properties, could one implement DPG-da? Or are there additional reasons why random forests are necessary?

**Audience:**

Yes

**Audience Explanation:**

Class imbalance is a widespread challenge so any broadly applicable method for correcting imbalance may be of interest to some readers.

**Broader Impact Concerns:**

I do not have any particular broader impact concerns, only those standard for all machine learning research apply.

**Claims And Evidence:**

Yes

**Claims Explanation:**

Technically, the core claims of the paper are only that they propose a new framework and include empirical evaluations. These facts are true, though there are significant elements of the empirical evaluation that are not included in the current manuscript.

**Requested Changes:**

Please see the Core Questions in the “Summary of Contributions” section.

It is difficult to evaluate the violation counts reported in Figure 3 without knowing how many augmented samples are generated for each dataset-method pair. Can the authors present a normalized version for ease of comparison?

It is my understanding that as presented, DPG-da is mainly applicable to tabular data, due to the form of decision predicates considered (coordinate-wise thresholds). It may be appropriate to state this more explicitly in the abstract and/or introduction.

Correct minor typos:
- Missing spaces before some parenthetical citations (e.g. “...may induce overfitting(Dixit et al. 2023)” on page 1.
- Page 5, “It leverages interpretable the evolutionary trajectory…”; “...we generate feature-level analysis that visualize how candidate solutions evolve…”

---

> ### Author Response · Authors · 2026-01-07
> **Response**
>
> We thank the reviewer for their thoughtful questions and constructive suggestions. We address each point below.
>
> (1) Do models trained with synthetic data outperform the surrogate model?
> Yes. The surrogate model is never used as a final predictor. It is trained only on the training split to extract decision predicates, while downstream classifiers are trained on augmented data and evaluated on a completely untouched test set. We have added appendix material comparing baseline (no augmentation) performance against DPG-da across augmentation levels, including a scatterplot that explicitly shows when augmentation improves or degrades performance.
>
> (2) Potential bias toward surrogate decision boundaries.
> We agree this is an important concern. To mitigate bias, the surrogate is trained on a holdout split, and downstream evaluation is performed on untouched test data. Additionally, feasibility under the surrogate is only one component of the GA objective: sparsity, locality (distance to the query sample), and diversity terms encourage local, varied perturbations rather than convergence toward surrogate boundaries. While we do not directly measure boundary convergence, the observed improvements over the baseline—together with reported precision and recall—suggest that downstream models are not merely inheriting surrogate artifacts.
>
> (3) Strength of axis-aligned constraints and possible extensions.
> Axis-aligned thresholds were chosen because they are the native predicates of tree ensembles, enabling simple predicate extraction, efficient feasibility checks, and interpretable distance-to-constraint terms. However, DPG-da is not fundamentally limited to this choice. The framework only requires explicit predicates and a way to evaluate violations (and optionally distances). Oblique hyperplanes or predicates in embedded feature spaces could be incorporated by modifying predicate extraction and distance computation, while leaving the evolutionary optimization unchanged.
>
> (4) Apparent convexity/connectedness of constraint sets.
> Random forests can induce unions of disjoint regions. The DPG representation does not assume connectedness: it supports multiple alternative predicate conjunctions (i.e., disjunctions of paths) leading to the same class. The constraint summaries shown in Table 5 are simplified projections for readability and may appear convex, while the underlying feasible set used during optimization can remain disconnected.
>
> (5) F1-scores used in the critical difference analysis (Figure 5).
> Figure 5 uses the average macro F1-score across all augmentation levels (15%, 30%, and 50%) for each dataset–method pair. We have clarified this aggregation procedure in the revised manuscript.
>
> (6) Integer and discrete feature constraints.
> As currently presented, DPG-da operates in continuous space and does not inherently enforce integer-valued or discrete constraints. This can be addressed with straightforward extensions, such as per-feature precision parameters and projection (rounding and clamping) after mutation or before fitness evaluation. Similar constrained operators can be applied to categorical or one-hot encoded features.
>
> (7) Necessity of random forests as surrogates.
> Random forests were chosen for their interpretability and efficient predicate evaluation, but they are not required. Any surrogate that provides explicit predicates defining class-consistent regions and supports efficient feasibility checks can be used (e.g., gradient-boosted trees, rule sets, or oblique-tree models). We have clarified this generality in the revised text.
>
> (8) Normalization of violation counts and applicability scope.
> We have added normalized violation rates to Figure 3 to facilitate comparison across datasets and methods. We also explicitly state that DPG-da is currently targeted at tabular data, due to its predicate-based formulation, and we have corrected the minor typographical issues noted by the reviewer.
>
> We believe these revisions clarify the scope, assumptions, and empirical behavior of DPG-da and strengthen the overall presentation. We appreciate your comments.

---

> > ### Comment · Reviewer_CePR · 2026-01-13
> >
> > I thank the authors for their thorough responses and their updates to the manuscript.
> >
> > I want to briefly clarify my request for comparison with the surrogate model. The authors write, "Do models trained with synthetic data outperform the surrogate model? Yes. The surrogate model is never used as a final predictor. It is trained only on the training split to extract decision predicates, while downstream classifiers are trained on augmented data and evaluated on a completely untouched test set."
> >
> > I am confused: have the authors compared final, downstream classification performance on augmented data \textit{by the surrogate model} with the performance of the other classifiers? The authors respond that classifiers trained on augmented data outperform the surrogate model, but then also write that the surrogate is never evaluated on the test data.
> >
> > I am interested in this comparison because DPG-da uses the decision boundaries learned by the surrogate model (from the training set) as a proxy for ground truth. The synthetic data generated during augmentation reflects these boundaries, and as the authors acknowledge one might reasonably hypothesis that this process means other classifiers, trained on the augmented samples, may be biased towards the decision boundaries learned by the surrogate model. Given this, it seems reasonable and important to evaluate the surrogate model's performance on the holdout data in order to verify that models trained on augmented data outperform the baseline surrogate. If this is not true, then it is not worth the process of generating augmented samples (which use the surrogate model as "ground truth") to train other classifiers.

---

> > > ### Author Response · Authors · 2026-01-14
> > > **Response**
> > >
> > > We thank the reviewer for the clarification and agree that this comparison is central to understanding the role of the surrogate model in DPG-da. We apologize that this was not stated explicitly enough in the previous revision.
> > >
> > > To clarify, all models, ncluding the surrogate random fores, are evaluated on the same untouched hold-out test set. The surrogate model is trained only on the training split and its test-set performance is reported as a baseline. Downstream classifiers are then trained on training data augmented using DPG-da (which relies on the surrogate only for constraint extraction) and are evaluated on the same test set.
> > >
> > > The surrogate model’s test performance is explicitly reported in the appendix (Table 9, RF at 0% augmentation), where it serves as a reference baseline. As the reviewer anticipated, and as our results confirm, the surrogate model achieves lower performance on average than classifiers trained on DPG-da–augmented data. This supports the motivation for using the surrogate as a constraint-extraction mechanism rather than as a final predictor.
> > >
> > > We note that, as expected in a heterogeneous benchmark setting, there are individual datasets for which the surrogate (or the non-augmented baseline) outperforms DPG-da. These cases are explicitly documented in Appendix F, including Table 7 and Figure 13. Importantly, however, the aggregate analysis across datasets and classifiers shows that models trained on DPG-da, augmented data outperform the surrogate on average, as reflected in the critical difference analysis and the overall ranking results.
> > >
> > > Taken together, these results demonstrate that DPG-da does not simply propagate or replicate the surrogate’s decision boundaries. Instead, while the surrogate provides a structured notion of feasibility, the augmented data enables downstream classifiers to achieve better generalization than the surrogate itself on unseen test data. We have revised the manuscript to make this comparison and its interpretation explicit.

---

> > > > ### Comment · Reviewer_CePR · 2026-01-14
> > > >
> > > > I thank the authors for directing me to Table 9; that's exactly the kind of data I was interested in.
> > > >
> > > > Comparing Table 8 and Table 9, I count that on 10 out of 27 benchmark datasets, the F1 score for RF at 0% is equal to or higher than the best-performing augmented classifier method, averaged over augmentation levels. I would suggest that in the future the authors perform some statistical tests evaluating to what extent classifiers trained on augmentation are outperforming the surrogate model across benchmarks. It's my impression Table 7 and Figure 13 compare the performance of classifiers (not necessarily random forrest/the surrogate model) at 0% augmentation with the performance of the same classifiers after augmentation. This is different than comparing the surrogate to the later classifiers.

---

### Review · Reviewer_jKRJ · 2026-01-03

**Summary Of Contributions:**

The paper proposes Decision Predicate Graphs for Data Augmentation (DPG-da), a novel over-sampling framework designed to address the problem of "infeasible" or semantically invalid samples often generated by traditional interpolation-based methods (e.g., SMOTE) and generative models. The core innovation involves training a surrogate ensemble model (Random Forest) to extract class-specific constraints via a Decision Predicate Graph (DPG). These constraints then guide a Genetic Algorithm (GA) to generate synthetic minority samples that strictly adhere to the feasible regions of the data.

Key Strengths:
* Methodological Rigor regarding Validity: The paper effectively highlights a critical flaw in existing literature—the generation of physically or logically impossible samples (e.g., negative blood pressure). The "Violation Analysis" on handcrafted and real-world datasets provides strong evidence that DPG-da eliminates such violations while baselines like SMOTE-LVQ frequently fail.
* Performance: Extensive benchmarking on 27 datasets demonstrates that DPG-da consistently outperforms 12 state-of-the-art baselines in terms of Macro F1-score, validated by Friedman and Nemenyi statistical tests.
* Traceability: The method offers a transparent view of the augmentation process by visualizing feature changes during the evolutionary steps (Section 6.3).

Key Weaknesses:
* Computational Efficiency: The method is orders of magnitude slower than standard baselines (roughly 1,809 seconds vs. $<1$ second), which significantly limits its scalability.
* Surrogate Dependence: The validity of the constraints relies entirely on the quality of the surrogate model's approximation, a limitation acknowledged in the Appendix but not fully explored in the main experiments.

**Audience:**

Yes

**Audience Explanation:**

The problem of handling imbalanced tabular data is ubiquitous in machine learning applications, particularly in high-stakes domains like healthcare and fraud detection where data validity is paramount.

**Claims And Evidence:**

Yes

**Claims Explanation:**

1. Realism/Feasibility: The authors provide a specific "Violation Benchmark" (Section 5.3) and use heatmaps (Figure 3) to quantify how often baselines break domain rules compared to DPG-da. The scatter plots in Figure 10 visually confirm that DPG-da respects feature boundaries where others fail.
2. Performance: The classification experiments are comprehensive, covering 27 UCI datasets with varying imbalance ratios. The authors employ appropriate statistical rigor, using the Friedman test ($p < 0.001$) and a Nemenyi post-hoc test to establish significant differences in rank.
3. Traceability: The claim of "traceability" is supported by the feature-wise change logs in Table 1 and Figure 7, showing exactly which features were modified during the Genetic Algorithm optimization.

**Requested Changes:**

1. Computational Efficiency and Scalability Analysis:
   The paper explicitly states that DPG-da requires approximately 1,809 seconds to run, compared to less than 1 second for most baselines (Figure 6). This is a discrepancy of several orders of magnitude.
    * Action: Please provide a more rigorous discussion on the time complexity of the method, specifically regarding the Genetic Algorithm component.
    * Action: Please discuss the scalability of DPG-da on datasets with higher dimensionality ($d > 100$) or larger sample sizes ($N > 100k$). Is the method practical for anything beyond small-to-medium tabular datasets? A brief "Limitations" discussion in the main text (moving key points from Appendix A) is necessary to set realistic expectations for practitioners.

2. Validation of Surrogate Model Quality:
   The validity of the generated constraints ($C$) relies entirely on the quality of the surrogate model (Random Forest). If the surrogate model fails to capture the true decision boundary (low fidelity), the "feasible regions" it defines might be incorrect, leading to valid samples being rejected or invalid ones being accepted.
    * Action: Please report the predictive performance (e.g., F1-score or Accuracy) of the surrogate model itself on the training data for the benchmark datasets. This is crucial to verify that the constraints extracted are grounded in a reliable approximation of the data distribution.

Would Strengthen the Work:

3. Refinement of "Interpretability" Terminology:
   The paper claims the method offers "interpretability" by showing the evolutionary trajectory of samples (Figure 7, 8). However, strictly speaking, this demonstrates traceability (knowing *how* a sample changed) rather than inherent interpretability (knowing *why* the model considers this valid vs. invalid in a causal sense).
    * Recommendation: I suggest refining the terminology in Section 4.3 to distinguish between "traceability of the generation process" and "interpretability of the data structure." This nuance would improve the conceptual precision of the paper.

4. Modern Generative Baselines:
   While the inclusion of CTGAN and TVAE is appreciated, the field of tabular generation has moved towards diffusion models (e.g., TabDDPM) in recent years.
    * Recommendation: Adding a brief comment or a small experiment comparing DPG-da against a diffusion-based baseline would better contextualize the work within the current SOTA landscape.

---

> ### Author Response · Authors · 2026-01-08
> **Response**
>
> We thank the reviewer for the thoughtful and constructive feedback. We have revised the manuscript to address these points and to set clearer expectations about the scope and limitations of DPG-da.
>
> Regarding computational efficiency and scalability, we fully acknowledge that DPG-da is substantially more expensive than classical over-sampling methods, as reflected in Figure 6. To address this concern, we have added a dedicated analysis in Appendix G that discusses the time complexity of the method. The evolutionary search, whose dominant cost scales with the number of fitness evaluations (population size × generations × number of augmented samples) and the cost of constraint checking, which depends on feature dimensionality and predicate count. We additionally provide an empirical analysis correlating runtime with dataset characteristics such as dimensionality and sample size.
>
> Following the reviewer’s suggestion, we have moved the limitations discussion from the appendix into the main text to clearly communicate these trade-offs to practitioners.
>
> Concerning the validity of the surrogate model, we agree that the quality of the extracted constraints depends on the fidelity of the surrogate. To address this, we now report the predictive performance of the surrogate random forest on the baseline (non-augmented) data in Section 6.2.1, and include a detailed table with surrogate performance across all benchmark datasets in Appendix F. These results show that the surrogate consistently achieves strong performance, supporting the reliability of the constraint regions used during augmentation.
>
> We also appreciate the reviewer’s comment on interpretability terminology. We have refined the terminology in Section 4.3 to distinguish between interpretability of the data structure (via explicit, predicate-based constraints) and traceability of the generation process (visualizing how candidate samples evolve during optimization). This clarification avoids overstating the interpretability claims while preserving the intended contribution.
>
> Finally, while a full experimental comparison with diffusion-based tabular generators is beyond the scope of this revision, we have added a brief discussion to contextualize DPG-da relative to recent diffusion-based approaches, highlighting the complementary nature of our constraint-driven, interpretable framework compared to distribution-learning generative models.
>
> We believe these revisions significantly strengthen the paper by clarifying scope, transparency and sharpening conceptual framings that were lacking.

---

### Decision · Action_Editor_rS92 · 2026-02-12

**Recommendation:** Reject

**Audience:**

Yes

**Audience Explanation:**

The problem of imbalanced tabular data is still of interest in machine learning research.

**Claims And Evidence:**

No

**Claims Explanation:**

While all reviewers acknowledge the relevance of the problem, points were raised on the methodological contribution, its complexity and dependence on the surrogate model. One of the reviewers raised a valid argument on the performance of the surrogate model compared to the best performance of the augmented classifiers. The reviewer argues that in 10 out of 27 of the studied benchmarks the simple random forest model outperforms the best of augmented classifiers. This important as it does not support authors claims of significant performance improvement but rather as a result of additional data augmentation merely. Given the computational cost of the method as raised by several reviewers and  the lack of theoretical justifications, the contribution of this paper remains limited.

**Resubmission Of Major Revision:**

The authors may consider submitting a major revision at a later time.